# Timing matters in macrophage/CD4+ T cell interactions: an agent-based model comparing *Mycobacterium tuberculosis* host-pathogen interactions between latently infected and naïve individuals

Alexis Hoerter,[1] Alexa Petrucciani,[1] Jordan Bonifacio,[2] Eusondia Arnett,[2] Larry S. Schlesinger,[2] Elsje Pienaar[1,3]

**ABSTRACT**   Tuberculosis (TB), caused by *Mycobacterium tuberculosis* (*Mtb*), remains a significant health challenge. Clinical manifestations of TB exist across a spectrum with a majority of infected individuals remaining asymptomatic, commonly referred to as latent TB infection (LTBI). *In vitro* models have demonstrated that cells from individuals with LTBI can better control *Mtb* growth and form granuloma-like structures more quickly, compared to cells from uninfected (*Mtb*-naïve) individuals. These *in vitro* results agree with animal and clinical evidence that LTBI protects, to some degree, against reinfection. However, the mechanisms by which LTBI might offer protection against reinfection remain unclear, and quantifying the relative contributions of multiple control mechanisms is challenging using experimental methods alone. To complement *in vitro* models, we have developed an *in silico* agent-based model to help elucidate host responses that might contribute to protection against reinfection. Our simulations indicate that earlier contact between macrophages and CD4+ T cells leads to LTBI simulations having more activated CD4+ T cells and, in turn, more activated infected macrophages, all of which contribute to a decreased bacterial load early on. Our simulations also demonstrate that granuloma-like structures support this early macrophage activation in LTBI simulations. We find that differences between LTBI and *Mtb*-naïve simulations are driven by TNFα and IFNγ-associated mechanisms as well as macrophage phagocytosis and killing mechanisms. Together, our simulations show how important the timing of the first interactions between innate and adaptive immune cells is, how this impacts infection progression, and why cells from LTBI individuals might be faster to respond to reinfection.

**IMPORTANCE**   Tuberculosis (TB) remains a significant global health challenge, with millions of new infections and deaths annually. Despite extensive research, the mechanisms by which latent TB infection (LTBI) confers protection against reinfection remain unclear. In this study, we developed an *in silico* agent-based model to simulate early immune responses to *Mycobacterium tuberculosis* infection based on experimental *in vitro* infection of human donor cells. Our simulations reveal that early interactions between macrophages and CD4+ T cells, driven by TNFα and IFNγ, are critical for bacterial control and granuloma formation in LTBI. These findings offer new insights into the immune processes involved in TB, which could inform the development of targeted vaccines and host-directed therapies. By integrating experimental data with computational predictions, our research provides a robust framework for understanding TB immunity and guiding future interventions to mitigate the global TB burden.

**KEYWORDS**   tuberculosis, agent based model, trained immunity, granuloma formation, host-cell interactions, host response, computational model

**Peer Reviewer** Evan Skowronski, US Army Edgewood Chemical Biological Center, Aberdeen Proving Ground, Maryland, USA

Address correspondence to Elsje Pienaar, epienaar@purdue.edu.

The authors declare no conflict of interest.

See the funding table on p. 25.

Tuberculosis (TB) is a highly prevalent disease, causing 10.6 million newly developed infections and 1.3 million deaths in 2022 (1). Clinical manifestations of TB exist across a broad spectrum with active TB disease on one end and bacterial clearance on the other with latent TB infection (LTBI) falling in between (2–4). LTBI is characterized as individuals having evidence of infection with *Mycobacterium tuberculosis* (*Mtb*), without any clinical signs or symptoms (4–6), and a quarter of the global population is estimated to be latently infected with *Mtb* (7, 8). There is a 5%–10% risk of LTBI progressing to active disease over the course of a lifetime (2, 9–12). However, the majority of LTBI individuals are considered to be long-term controllers, i.e., remain clinically asymptomatic.

The extent to which LTBI might confer protection and reduce the risk of progressive disease after reinfection in adults remains unclear. It is suggested that LTBI individuals have a 35%–81% lower risk of progressing to active TB after reinfection compared with uninfected individuals who are infected for the first time (13–17). Furthermore, animal models of *Mtb* infection conferred strong protection against reinfection with *Mtb* as demonstrated by lower bacterial burden compared with primary infection (18–20). *In vitro* studies also showed that cells from LTBI or purified protein derivative positive (PPD+) individuals control bacterial growth better than cells from uninfected (PPD−) individuals (21–25). In particular, Guirado et al. (26) showed that peripheral immune cells from LTBI individuals can better control bacterial growth *in vitro*, have increased proliferative activity, form granuloma-like structures faster, and have higher levels of inflammatory cytokines compared to *Mtb*-naïve individuals (26). However, other *in vitro* studies have shown worse bacterial control by LTBI individuals compared to uninfected individuals (27–29). These differing outcomes could be due to population or geographic variation (low vs high TB endemic area) of the study cohorts or the method used for measuring bacterial growth. Despite these contrasting data, it is clear that host immunological differences resulting from LTBI have implications for infection progression.

In this work, we assess the impact of early host-pathogen interaction events on *Mtb* infection progression. In particular, we investigate the impact of initial events associated with early granuloma formation in LTBI vs naïve individuals. Granulomas are unique microenvironments orchestrated by the immune response to contain *Mtb* and localize host-pathogen interactions. Early events associated with granuloma development in *Mtb* infection have been shown to impact late-stage clinical outcomes (30–32). However, studying the initial infection dynamics that lead to granuloma formation is challenging *in vivo*. To address this challenge, *in vitro* models have successfully (26, 33–35) captured key characteristics of *Mtb* within the granuloma microenvironment using peripheral blood mononuclear cells (PBMCs), lung-on-a-chip models, bioelectrospray microspheres, and collagen matrices. In this work, we characterize an *in vitro* model that investigated how host infection status (LTBI vs *Mtb*-naïve) affects bacterial control and the formation of granuloma-like structures (26). While this *in vitro* model showed that cells from LTBI individuals are better able to control *Mtb* growth, the underlying mechanisms driving the observed differences remained unclear. Here, we use computational modeling to address these questions.

Within-host mathematical and computational mechanistic models of TB have been used to investigate antibiotic treatment, vaccine development, biomarker discovery, cytokine and chemokine dynamics, and bacterial dynamics (reviewed in references [36, 37]). Agent-based models (ABMs) are particularly useful for connecting molecular-level interactions to cellular and tissue level outcomes and for providing robust spatio-temporal information about *Mtb* infection progression. Here, we apply well-established agent-based modeling techniques to simulate an *in vitro* system (26) and characterize mechanistic differences between LTBI and *Mtb*-naive individuals. We simulate *in vitro* *Mtb*-immune interactions for human donor cells from LTBI and *Mtb*-naïve individuals (referred to as "naïve").

## MATERIALS AND METHODS

### Model description

Our ABM is based on *in vitro* granuloma structures generated through *Mtb* infection of human donor PBMCs from LTBI and *Mtb*-naïve donors, as published in reference (26). Briefly, in the *in vitro* model, PBMCs are purified from individuals with a positive Mantoux screening test and/or interferon-gamma release assay (IGRA) within the previous 12 months indicating LTBI and from negative test result individuals (naïve). A total of $2 \times 10^6$ PBMCs/mL are added to either 24- or 96-well plates, with $2 \times 10^5$ monocytes/mL (10% of PBMCs) infected at a 1:1 multiplicity of infection (MOI) with *Mtb*, while the remaining cells are classified as lymphocytes. PBMCs and *Mtb* are cultured for 8 days. Bacterial growth was measured in colony forming units (CFU) per milliliter, the proliferative activity of cells in granuloma-like structures was determined by a naphthol blue black-based cell enumeration assay using phase-contrast microscopy, and granuloma-like structure development in cultures was assessed via light microscopy. A granuloma scoring index, determined by the size of the structure, was used to measure granuloma-like structure formation. Concentrations for several cytokines in the supernatants, including IFNγ and TNFα, were determined by enzyme-linked immunosorbent assay (ELISA) (26).

We emulate this *in vitro* experimental system in our computational model. The ABM contains two main agent classes—immune cells and bacteria—that interact in a 3D environment composed of a rectangular grid that represents a subsection of the *in vitro* culture well. Immune cells are further divided into macrophages and CD4+ T cells. We also simulate the secretion, diffusion, and function of two cytokines, TNFα and IFNγ. Time is discretized into 6-minute time steps in which agents are selected in a random order to follow its set of rules. This time step was chosen because it is the approximate time for a macrophage to move 20 µm (its own length) (38–42) or one grid compartment. The simulation is run for a total of 8 days to mimic the *in vitro* experimental time frame. A schematic of the agents and rules is depicted in Fig. 1 (model mechanisms are described below with parameter names italicized and parameter values in Table 2 and Table S1 at https://doi.org/10.5281/zenodo.13844841). Model mechanisms are drawn in part from a well-established ABM of non-human primate (NHP) granulomas, *GranSim* (43–49). The ABM presented here has the same model structure as Petrucciani et al. (50, 51) but with unique parameters and initial conditions described here to reflect the PBMC *in vitro* model under consideration in this work. The simulation is implemented in Java using Repast Simphony 2.8 (52). Simulations were run on Purdue Brown Cluster, XSEDE resources, and ACCESS resources. MATLAB and Python are used for data analysis and visualization. ChatGPT was used to create and edit graphing scripts for MATLAB.

### Environment

Our ABM consists of a 3D grid measuring $100 \times 100 \times 10$ grid compartments where each grid compartment is 20 µm $\times$ 20 µm $\times$ 20 µm (approximately the size of a macrophage) for a total space dimension of 2 mm $\times$ 2 mm $\times$ 0.2 mm. This 3D grid represents a small subsection of one well in a 24- or 96-well plate with the bottom layer of the grid representing the bottom of the well. We assume that this section of one well is similar to neighboring sections and therefore assume toroidal boundaries in the x-y directions and no flux boundaries in the z direction (Fig. 1a). Each grid space allows single occupancy for immune cells but permits multi-occupancy for bacteria due to the size discrepancy between the two (~20 µm [53] vs ~1–10 µm long [54], respectively).

### Cytokines

There are two cytokines represented in our simulation: TNFα and IFNγ. These cytokines are represented as continuous variables in the grid. Cytokines are secreted from immune cells and diffuse throughout the grid based on their diffusion coefficients and also degrade at some rate. Diffusion is modeled using a three-dimensional version of the alternating-direction explicit (ADE) method, where a spatial step represents a grid

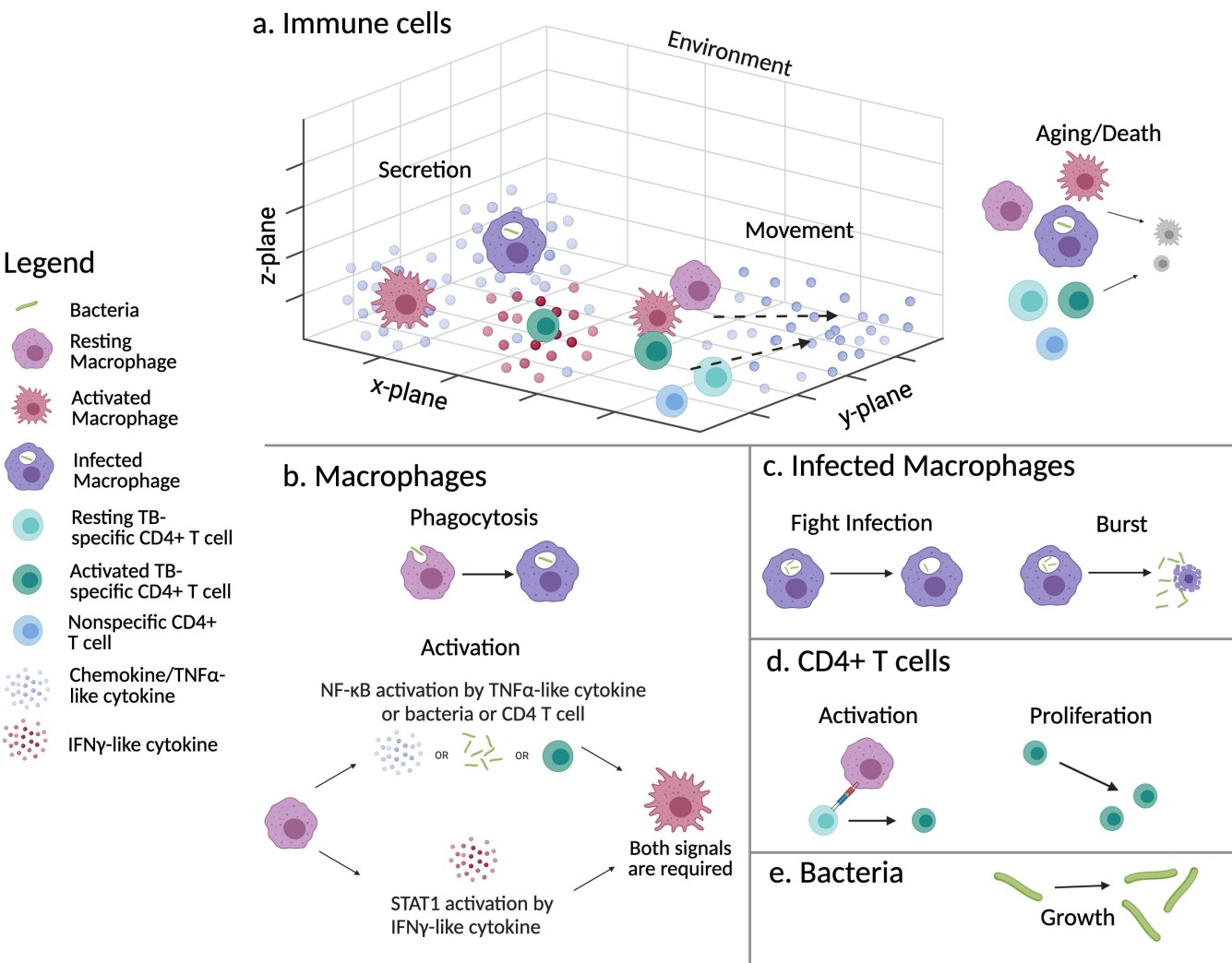

**FIG 1** Schematic of ABM agents and rules adapted from Petrucciani et al. (51). (a) All immune cells and bacteria exist in a 3D environment and remain at the bottom z-layer unless on top of another cell. Activated CD4+ T cells secrete TNFα and IFNγ, activated macrophages secrete TNFα, and infected macrophages secrete TNFα. All immune cells except infected macrophages move probabilistically along the TNFα concentration gradient. All immune cells have ages (resting and activated) and die when their maximum lifespan is reached. (b) Macrophages can phagocytose *Mtb*. Macrophages can become activated by activating both NF-κB and STAT1 pathways. (c) Infected macrophages can attempt to kill intracellular *Mtb*, but if unsuccessful will burst when intracellular bacteria reach a threshold, releasing *Mtb* into the environment and die. (d) TB-specific CD4+ T cells can become activated and can divide. (e) *Mtb* will grow either extracellularly or intracellularly. Created in Biorender.

compartment (55, 56). The ADE method can have a maximum Δ*t* of approximately 4–6 times the Δ*t* predicted by the conditional stability criterion of the forward-time central-space method to maintain acceptable accuracy (55); therefore, we have incorporated a parameter (*diffusionTimeStepMultiplier*) to the diffusion time step to reduce run time. Additionally, to represent slower diffusion due to cellular crowding, diffusion through a granuloma is allowed to be some fraction from 0 to 1 (*granulomaFractionOfDiffusion*) of the sampled diffusion coefficient.

### Immune cells

All immune cells (except for infected macrophages) move about the environment by chemotaxis along the concentration gradient of TNFα and based on a probability parameter that is associated with their functional status (resting or activated). For the sake of computational efficiency (which is greatly affected by each additional diffusing molecule), we chose to assume a pseudo-steady state relationship between TNFα and

its downstream chemokines, thereby allowing us to use TNFα as a proxy for chemokines. Movement of all activated immune cells is varied from 0% to 50% (probability of movement per time step), while resting movement is varied from 50% to 100% (100% represents the scenario where cells are allowed to move every time step) based on the assumption that activated cells would conserve energy by moving slower (57). If a random number is less than the movement probability then a cell will attempt to move. All the possible grid compartments in the Moore neighborhood (26 grid compartments immediately surrounding the current grid compartment plus its current location = 27 grid compartments to evaluate) are compiled as eligible grid compartments to move to with the exception that the cell cannot consider a "floating" location. Cells can move upward in the z-dimension by crawling on top of another immune cell but cannot move to a grid compartment that would result in the cell "floating" (Fig. 1a). After removing these "floating" locations, the values of the TNFα concentration for each non-floating grid compartment in the cell's Moore neighborhood are summed. We calculate the probability of a cell moving to a specific grid compartment in its Moore neighborhood by dividing the concentration of TNFα for that grid compartment by the total TNFα concentration for the eligible non-floating grid compartments in the Moore neighborhood. If the total concentration of TNFα in the surrounding Moore neighborhood is smaller than the concentration threshold for movement (*tnfThresholdForImmuneCellMovement*), then the cytokine signal is too small to warrant anything besides random movement, and the cell will choose a random direction to move. Cells can only move one grid compartment per time step. If the cell moved on top of another cell and the bottom cell moved away in the same time step, a check at the end of the time step would induce "gravity" meaning that cells fall in the z-dimension if no other immune cell is below them. All immune cells age according to an individualized lifespan that accelerates when the cell is activated (51).

### Macrophages

In the simulation, macrophages can be classified as resting or activated as well as uninfected or infected. The simulation is initialized with resting, uninfected macrophages uniformly and randomly distributed throughout the grid. Macrophages follow the movement and aging rules described above in "Immune cells." All macrophages are initialized with unique activation spans for both the activation pathways (*macrophageNFkbSpan* and *macrophageStat1Span*). Resting healthy macrophages are uninfected macrophages that can phagocytose *Mtb* within the environment as they move around. A macrophage will create a list of all the *Mtb* within its Moore neighborhood and randomly choose one of the list to attempt to phagocytose if it is extracellular. A resting uninfected macrophage will phagocytose at *basePhagocytosisProbability*. If phagocytosis is successful, then this is considered an infection event, and this uninfected macrophage will become an infected macrophage.

### Infection status

As previously mentioned in "Immune cells," infected macrophages no longer move throughout the environment. They may continue to phagocytose bacteria that are in their Moore neighborhood as long as the number of intracellular *Mtb* is below the *phagocytosisThreshold*. If the number of intracellular *Mtb* is above this value, the infected macrophage will no longer continue to phagocytose *Mtb*. Each time step infected macrophages have some probability to kill its intracellular *Mtb* based on its activation status, depending on if the infected macrophage is activated or not. If all intracellular *Mtb* are killed, the infected macrophage will become an uninfected macrophage keeping its status as either resting or activated. If the number of intracellular *Mtb* exceeds a threshold, the infected macrophage will lose its functional capability to kill intracellular *Mtb* and cannot become fully activated. Once the number of intracellular *Mtb* is greater than the *burstLimit*, the infected macrophage dies and is removed from the simulation, and all intracellular *Mtb* are put on the grid and become extracellular *Mtb*.

Each individual macrophage is assigned a random *burstLimit* between 20 and 40 (58). If an infected macrophage reaches the end of its lifespan, the intracellular *Mtb* will be released back into the environment similar to bursting. Infected macrophages secrete TNFα at a baseline level of *infectedMacrophageTNFSecretion* molecules/cell/s but if the infected macrophage is also activated, it will secrete at a rate of twice the *activatedMacrophageTNFSecretion* molecules/cell/s (59, 60).

### Activation status

Infected or uninfected macrophages can become activated. Once a macrophage is activated, it will utilize the activated phagocytosis, activated killing probability, and activated TNFα secretion rate. Activated infected macrophages have a greater probability of killing intracellular *Mtb* compared to infected macrophages that are not activated. Two pathways, NF-κB and STAT1, are required for a macrophage to become fully activated. The NF-κB pathway can be activated one of three ways: extracellular *Mtb* in the Moore neighborhood above a threshold (*bacThresholdForNFkBActivation*), TNFα concentration at that grid compartment above a threshold (*tnfThresholdForNFkBActivation*), or randomly pulling an activated CD4+ T cell off the list of all the CD4+ T cell within the Moore neighborhood. These represent activation of toll-like receptors (TLR), TNFα interaction with tumor necrosis factor receptor (TNFR), and CD40–CD40L interactions, respectively (48, 61–63). All three of these potential NF-κB pathways will be checked in a random order. The STAT1 pathway is only activated by a concentration of IFNγ above a threshold (*ifnThresholdForStat1Activation*) and is a proxy for macrophage polarization toward a more proinflammatory phenotype (48). NF-κB and STAT1 activation each last for a set duration (*macrophageNFkbSpan* and *macrophageStat1Span*). These durations have variance added to introduce heterogeneity into the population. After the set duration for each pathway is exceeded, the macrophage will deactivate with the ability to reactivate again if the required conditions are met. If both pathways are activated at the same time, the macrophage is considered fully activated.

### CD4+ T cells

There are two groups of CD4+ T cells in this simulation: non-TB-specific and TB-specific CD4+ T cells. Both types follow the movement and aging rules described above in "Immune cells." The difference between the two is that only TB-specific CD4+ T cells can become activated and proliferate. TB-specific CD4+ T cells can become activated by putting all the macrophages in its Moore neighborhood on a list and randomly choosing one off the list. If the randomly chosen macrophage has interacted with *Mtb* (i.e., the macrophage is either an infected macrophage or an uninfected macrophage that successfully killed all intracellular *Mtb*), then the CD4+ T cell can become activated with some probability (*cd4ActivationProbability*). This simulation mechanism represents antigen presentation on MHC II (64). To indirectly account for anti-inflammatory feedback, activated CD4+ T cells have a limited activated lifespan (65–67) and also have a probability of deactivating (*cd4DeactivationProbability*) each timestep. Activated CD4+ T cells can also divide with a doubling time of *cd4PopulationDoublingTime\** (1 ± *cd4DoublingTimeVariance*) until the maximum number of generations is reached (*maximumCD4Generations*). Activated CD4+ T cells secrete IFNγ and TNFα at rates of *activatedCD4IFNSecretion* and *activatedCD4TNFSecretion*, respectively, in molecules/cell/s (59, 60).

### Mycobacterium tuberculosis

*Mtb* is uniformly randomly distributed throughout the grid upon initialization as extracellular bacteria within the center 81% of the compartments to minimize edge effects in the simulation grid. Intracellular and extracellular growth rates of *Mtb* are defined as follows:

$$Mtb \text{ growth rate} = \frac{\ln(2)}{\text{doubling time (h)}}$$

where doubling time refers to either the intracellular (*mtbInternalDoublingTime*) or extracellular (*mtbExternalDoublingTime*) doubling time. A variance percentage (*mtbGrowthRateVariance*) is added or subtracted from the baseline doubling time to allow for diversity within the bacterial population. It is known that *Mtb* can adapt to its environment to replicate more slowly during infection (68). However, due to the complexity of this mechanism and the length of simulated infection (only 8 days), adjustable growth rates according to environmental signals or gene signatures were not incorporated in this simulation. Based on the *in vitro* experimental data, it is unknown whether the lower CFU per milliliter in LTBI can be attributed to growth inhibition via slower growth rates or bacterial killing by macrophages. Therefore, for this simulation, we computationally represent all growth inhibition as bacterial killing rates. Each individual *Mtb* starting biomass is randomly assigned between 0.5 and 1.5, and the biomass is updated every time step as follows:

$$Mtb \text{ biomass } (t_2) = \text{biomass}(t_1) * (1 + \text{growth rate})$$

Once a bacterium's biomass reaches the *divisionBiomassThreshold*, it undergoes division. Bacterial division is asymmetric (69). Thus, the newly divided bacterium's biomass is calculated as follows:

$$\text{New biomass} = \frac{\text{parent bacteria biomass}}{2} * \text{asymmetricDivisionPercentage}$$

Where the *asymmetricDivisionPercentage* is a random percentage between $1 \pm$ the *divisionBiomassVariance* (which is estimated at 20%) (69). After dividing, the new biomass is subtracted from the parent bacteria, and new growth rates are sampled. New extracellular bacteria are spatially offset by a random angle in the *x*-, *y*- and *z*-dimension using a radius between 0.1 and *newMtbPlacementRange*.

### Granuloma definition

In our ABM, granuloma formation is an emergent property of the simulation, and granulomas are defined as a cluster of cells grouped together without grid squares between them. The diagram in Fig. S1 at https://doi.org/10.5281/zenodo.13844841 illustrates how a cell determines if it is considered in a granuloma, which is done for every agent at every time step. Briefly, if the cell does not belong to a granuloma already, the cell surveys the number of cells in the surrounding Moore neighborhood. If the number of immune cells immediately surrounding the cell doing the check is ≥5 (*cellsNeededToBeAddedToGran*), then an additional check is done to see if any of those neighboring cells already belong to a granuloma. If any of those cells belong to a granuloma, then the cell doing the check is added to the granuloma with the closest center location. If no granulomas are found, then the cell checks to see if it meets the qualifications to form a new granuloma. If the number of cells is ≥12 (*granQualificationImmuneCellCount*), a new granuloma is formed with this cell and its surrounding cells. These parameter values (minimum of 5 for addition and 12 for new granuloma) were determined through trial and error based on visual inspection of structures that had acceptable gaps between cells to be considered a cluster. We track individual granulomas over time by assigning each an ID number.

If a cell is already associated with a granuloma, there are checks to determine whether the cell should still be considered a part of the granuloma. This is necessary as it is possible that granulomas disaggregate or individual cells move away from a granuloma. Again, all the immune cells in the Moore neighborhood are checked. If the number of immune cells surrounding this cell is ≥5 (*cellsNeededToRemainInGran*), then all the

granuloma ID numbers of those cells are determined. If the granuloma ID numbers match to that of the cell doing the check, then no changes are required. If the number of surrounding immune cells does not meet the requirement of *cellsNeededToRemainInGran* or the granuloma ID number of its neighbors does not match that of the cell being checked, then this cell is removed from the granuloma. Additionally, there is a check to see if a granuloma contains the minimum number of cells to qualify it for granuloma status. If there are fewer than 12 cells, then that granuloma fails to meet the minimum cell requirement and should no longer exist. It is then cleared from the list of granulomas currently in the simulation.

### Initial conditions and first estimate parameter ranges

The initial concentration of macrophages, CD4+ T cells, and *Mtb* are calculated from the *in vitro* experimental setup (26) and described in Table 1. Due to computational limitations, we cannot simulate the entire well but rather define our simulation grid (0.2 cm × 0.2 cm × 0.02 cm = 0.0008 cm$^3$) first. For this volume of our simulation, we multiplied the experimental concentration ($2 \times 10^6$ PBMCs/mL) by our simulation volume. Therefore, we initialize the simulation with 1,600 PBMCs (160 macrophages and 1,440 lymphocytes). This follows the assumption that PBMCs can be generally divided into 90% lymphocytes and 10% monocytes (26). Since we are only concerned with simulating CD4+ T cells, we determined from the literature that 50%–85% of lymphocytes are comprised of CD3+ T cells and that 30%–79% of these CD3+ T cells are CD4+ T cells (70–72). Furthermore, we must define what fraction of these CD4+ T cells are TB specific which we estimate to be 0.005%–0.5% of CD4+ T cells (73), the minimum chosen at 0.005% to have at least 1 TB-specific CD4+ T cell in a simulation with the remaining to be considered non-TB specific. We assume that naïve samples have some TB-specific cells that are naïve (i.e., have not encountered their cognate antigen yet and are therefore not effector cells) and are part of the diverse repertoire of T-cell receptors (TCRs) capable of responding to novel antigens throughout a human's life (74). To emulate the experimental MOI of 1:1 macrophage to *Mtb*, we initially added 160 bacteria to the simulation. The number of macrophages and bacteria added to the simulations are the only experimentally derived initial conditions. Table 1 summarizes the ranges of initial conditions which are the same for LTBI and naïve groups, reflecting the identical experimental procedures for LTBI vs naïve cells. All other parameters that impact initial conditions (fractionCD3, fractionCD4, and fractionTBSpecific) do not have experimental data available and are therefore estimated through calibration but constrained within biologically feasible ranges.

All other parameters are either determined by literature or estimated through calibration. Parameter ranges used at the start of our calibration were determined based on available literature or defined based on biological feasibility if no literature values could be determined (Table 2). Parameters that were not varied as part of calibration were determined through known literature values or estimated to be biologically feasible values (e.g., parameters related to variance) are shown in Table S1 at https://doi.org/10.5281/zenodo.13844841.

### Calibration

Our simulation is calibrated to published data from LTBI and naïve donor cells from Guirado et al. (26). Simulation outputs are calibrated to meet the following criteria: fall within (i) 0.5×–1.5× the experimental intracellular bacterial fold change at 3, 4, 5, 7, and 8 days post-infection, (ii) 0.6×–1.8× the experimental total cell fold change at day 7 post-infection, and (iii) day of first granuloma formation (day 3 or 4 post-infection for LTBI

**TABLE 1** Initial condition values and ranges for host cells and bacteria in the ABM$^a$

| Bacteria | Macrophages | Lymphocytes | Total CD4+ T cells | Number of TB-specific CD4+ T cells | Number of non-specific CD4+ T cells |
|---|---|---|---|---|---|
| 160 | 160 | 1440 | 216–966 | 1–48 | 215–918 |

$^a$These ranges are used in both LTBI and naïve simulations to reflect the identical experimental procedures for LTBI vs naïve cells.

**TABLE 2** Parameters that were varied in the simulations[a]

| Parameter name | Initial range | Unit | Reference(s) |
|---|---|---|---|
| **Bacteria** | | | |
| *mtbExternalDoublingTime* | 24–48 | Hours | (58, 75) |
| *mtbInternalDoublingTime* | 24–48 | Hours | (58, 76() |
| **Macrophages** | | | |
| *baseKillingProbability* | 0.0001–0.001 | Per tick | e |
| *activeKillingProbability* | 0.0011–0.01 | Per tick | e |
| *basePhagocytosisProbability* | 0.001–0.5 | Per tick | e |
| *activePhagocytosisProbability* | 0.51–1 | Per tick | e |
| *phagocytosisThreshold* | 1–15 | Internal bacteria | (61) |
| *cellularDysfunctionThreshold* | 12–25 | Internal bacteria | (61) |
| *nfkbSpan* | 0.11–115 | Hours | (48) |
| *thresholdForNFkBActivationTNF* | 5–500 | Molecules | e |
| *thresholdForNFkBActivationBac* | 20–500 | External bacteria | (61) |
| *stat1Span* | 0.11–115 | Hours | (48) |
| *thresholdForStat1ActivationIFN* | 5–500 | Molecules | e |
| *activatedMacrophageTNFSecretion* | 1–80 | Molecules/cell/s | (59, 60) |
| *infectedMacrophageTNFSecretion* | 1–80 | Molecules/cell/s | (59, 60) |
| *macrophagePopulation_MaxActivatedLifespan* | 5–25 | Days | (61) |
| *baseMovementProbabilityMacro* | 0.51–1 | Per tick | (38, 40, 41) |
| *activatedMovementProbabilityMacro* | 0–0.5 | Per tick | (57) |
| **CD4+ T cells** | | | |
| *fractionCD3* | 0.5–0.85 | CD3+ T cells/lymphocytes | (70–72) |
| *fractionCD4* | 0.3–0.79 | CD4+ T cells/CD3+ T cells | (70–72) |
| *fractionTBSpecific* | 0.005–0.05 | TB-specific CD4+ T cells/CD4+ T cells | (73) |
| *activationProbabilityCD4* | 0–1 | Per interaction | e |
| *deactivationProbabilityCD4* | 0.01–0.5 | Per tick | e |
| *activatedCD4TNFSecretion* | 1–80 | Molecules/cell/s | (59, 60) |
| *activatedCD4IFNSecretion* | 1–80 | Molecules/cell/s | (59, 60) |
| *cd4PopulationDoublingTime* | 4–16 | Hours | (65, 66) |
| *maximumCD4Generations* | 4–12 | Generations | (65, 66, 77) |
| *cd4PopulationMaxLifespan* | 34–340 | Days | (78) |
| *cd4PopulationActivatedLifespan* | 3–4 | Days | (65–67) |
| *baseMovementProbabilityCD4* | 0.51–1 | Per tick | e |
| *activatedMovementProbabilityCD4* | 0–0.5 | Per tick | e |
| **Diffusion** | | | |
| *thresholdForImmuneCellMovementTNF* | 5–500 | Molecules | e |
| *diffusionCoefficientTNF* | 0.1–1 e–7 | cm$^2$/s | (47, 49, 79–82) |
| *degradationRatePerSecondTNF* | 0.96–10 e–4 | s$^{-1}$ | (47, 49, 79, 80, 83) |
| *diffusionCoefficientIFN* | 0.1–1 e–7 | cm$^2$/s | (47, 49, 79–82) |
| *degradationRatePerSecondIFN* | 0.96–10 e–4 | s$^{-1}$ | ( 47, 49, 79, 80, 83) |
| *granulomaFractionOfDiffusion* | 0–1 | | e |

[a]Parameter ranges were determined by literature or estimated through preliminary stimulations (denoted by e). All parameters are continuous except *thresholdForNFk-BActivationBac*, *mtbInternalDoublingTime*, *mtbExternalDoublingTime*, *phagocytosisThreshold*, *cellularDysfunctionThreshold*, and *maximumCD4Generations*, which are discrete integer values. All parameters were sampled by a uniform distribution, except *nfkbSpan* and *stat1Span*, which were sampled by a log distribution. The initial range column represents the ranges that were used in the first iteration of the calibration.

and day 5 or 6 post-infection for naïve). We used WebPlotDigitizer (https://apps.autome-ris.io/wpd/) to identify values for CFU per milliliter from Fig. 2b in reference 26. We calculated the experimental fold change in CFU per milliliter for days 3, 4, 5, 7, and 8 relative to day 1 experimental value from this graph. In the simulations, fold changes for intracellular bacteria were calculated based on 4 hours post-infection to account for confounding effects of early dynamics in the simulations that were not explicitly

measured in the experiments. The mean total cell fold change value was given by Guirado et al. (26) for day 7 relative to initial values. For simulations, fold changes for total cell counts were calculated based on the initial number of cells. We use Latin hypercube sampling (LHS) with a centered design along with the alternating density subtraction (ADS) (84 ) method to perform iterative calibrations to identify a robust parameter region in which at least 75% of our parameter sets pass our criteria. We calibrate parameters for both LTBI and naïve data sets independently after the first LHS-ADS calibration iteration. All iterations are done with 2,500 parameter sets and three replicates each for a total of 7,500 runs. Calibration was complete after five iterations with 88.9% and 84% of parameter sets passing our criteria for LTBI and naïve data sets, respectively. A parameter set is considered to pass the three calibration criteria (i, ii, and iii above) if at least one of its replicates meets the criteria. The final calibrated parameter sets are available on Zenodo (see https://doi.org/10.5281/zenodo.13844841). We average the replicates for each final passing parameter set in our analysis.

## Sensitivity analysis

An LHS-partial rank correlation coefficients (PRCC) approach is used to determine parameter sensitivity for the LTBI and naïve parameter spaces identified from calibration (85). The median value of each parameter is determined for both LTBI and naïve calibrated parameter ranges. For the sensitivity analysis, parameters are varied 1 order of magnitude up and down from this median value, and the ranges for each parameter used in sensitivity analysis are shown in Table S2 at https://doi.org/10.5281/zenodo.13844841. If the parameter is a probability (meaning it can only be represented by a number from 0 to 1) and if the change in the maximum by increasing by 1 order of magnitude results in the value of the parameter going above 1, then it is capped at 1. We started with 500 LHS samples and increased the number of samples by 500 until the unique number of new parameters that had a significant PRCC value for four or more time points is less than 10% of the 37 varied parameters for both groups. (Note: if a parameter is newly significant for multiple outputs, it is only counted once.) This process indicated that using 2,000 samples for each group (LTBI and naïve) with three replicates each gives consistent results. These replicates are averaged before PRCCs are calculated. A significance level of 0.01 is used with a Bonferroni correction for the number of tests run (alpha = 1.365e−06). The relationship between the 37 varied parameters and outputs of interest was analyzed.

## Reproducibility

Given the parameter region that resulted in 88.9% of LTBI and 84% of naïve parameter sets passing, these parameter spaces are run with 2,500 samples and three replicates each twice more (producing all new combinations of parameters within these ranges) to test the robustness of the identified parameter region. For each of the resultant reproducibility batch runs, 88.6% and 88.8% parameter sets passed the LTBI criteria and 84.9% and 83.3% parameter sets passed for the naïve criteria. This indicates that the identified parameter region is reproducible.

## Statistical analysis

For all aggregate and granuloma level model output comparisons, a two-tailed *t*-test was initially performed with a significance level of 0.01 with a Bonferroni correction for the number of tests run (i.e., number of days in time course for the output of interest, eight for full infection comparison or four for granuloma comparisons resulting in α = 0.00125 for 8 day comparisons, and α = 0.0025 for 4 day comparisons). Even with this correction, the large number of samples makes this statistical test overpowered, and many distributions between LTBI and naïve look similar. Similarly, for parameter distribution comparisons, a two-tailed *t*-test with an alpha value of 0.01 identified 36 of the 37 varied parameters as significantly different between LTBI and naïve simulation parameters.

However, many of these parameter distributions look nearly identical with 18 of them having a percent difference in their means less than 10% between LTBI and naïve. Therefore, we applied the A-measure of stochastic superiority to determine which output and parameter distributions have a large deviation from stochastic equality (86). The A-measure of stochastic superiority is a generalization of the common language statistics that describes the probability that a random sample from one distribution is greater than a random sample from another distribution and can be used for continuous or discrete distributions (86). We followed the sample code provided by Hamis et al. (86) and used the scaled A-measure values with a threshold of 0.71 for stochastic inequality making the difference between the two distributions meaningfully significant. Many output comparisons were considered meaningfully significant; therefore, only non-significance is shown in figures. From the A-measure analyses, 26 parameters were determined to be stochastically different between LTBI and naïve parameter distributions. From this list of 26 parameters, we exclude the *bacThresholdForNFkBActivation* as no macrophages were ever NF-κB activated by the extracellular number of *Mtb*. These parameter differences are discussed below in TNFα and IFNγ related parameters along with bacterial killing probability drive increased bacterial killing in LTBI simulations.

## Experimental validation

To test model-predicted differences in intracellular vs extracellular bacterial populations, *in vitro* TB granuloma structures were generated as described elsewhere (26, 87–89). Briefly, human peripheral blood was collected from healthy Mantoux tuberculin skin test-negative individuals following Texas Biomed-approved Institutional Review Board protocols. All donors for these studies provided informed written consent. Heparinized blood was layered on a Ficoll-Paque cushion (GE Healthcare, Uppsala, Sweden) to allow for the collection of PBMCs. PBMCs were immediately infected with single cell suspensions of *Mtb* $H_{37}R_v$ (*Mtb* was obtained from the American Tissue Culture Collection [ATCC, Manassas, VA]) at MOI 1. The bacteria concentration and degree of clumping (<10%) were determined with a Petroff-Hausser Chamber. This method results in ≥90% viable bacteria, as determined by CFU assay. PBMCs + bacteria were incubated in RPMI with 10% autologous serum at 37°C/5% $CO_2$, and serum was replenished after 4 days. To assess the percentage of intracellular vs extracellular bacteria, the supernatant was removed and centrifuged at 100 g for 10 min, and cold DNase (Millipore-Sigma, Burlington, MA) was added to the adherent PBMCs. Centrifugation at 100 g was done to pellet non-adherent, viable PBMCs while leaving extracellular *Mtb* in the supernatant. These extracellular bacteria were diluted and plated on 7H11 agar (Remel, San Diego, CA). The pelleted non-adherent PBMCs were resuspended in 7H9 broth (BD Biosciences, Franklin Lakes, NJ) then added back to the wells containing DNAse, adherent PBMCs, and intracellular *Mtb*. Granuloma structures were then lysed with 0.05% SDS (FisherScientific) for 10 min, and then bovine serum albumin (BSA, ThermoFisher) was added to a final concentration of 4.3%. Lysates were diluted and plated on 7H11 agar. The number of CFUs was enumerated after growth for 3–4 weeks at 37°C.

## RESULTS

### Calibrated and validated simulation mirrors *in vitro* LTBI and naïve experimental group outcomes, predicting lower bacterial fold change, more proliferative activity, larger granuloma-like structures, and increased cytokine levels in LTBI

Through our calibration and validation process, we demonstrate that the simulations reproduce key trends and features of the experimental data in reference 26 for both LTBI and naïve cultures. Our ABM, calibrated independently to LTBI and naïve experimental group data, reproduces the range of experimental data for bacterial fold change (Fig. 2a) and total cell fold change (Fig. 2b) for both LTBI and naïve data sets. As in the

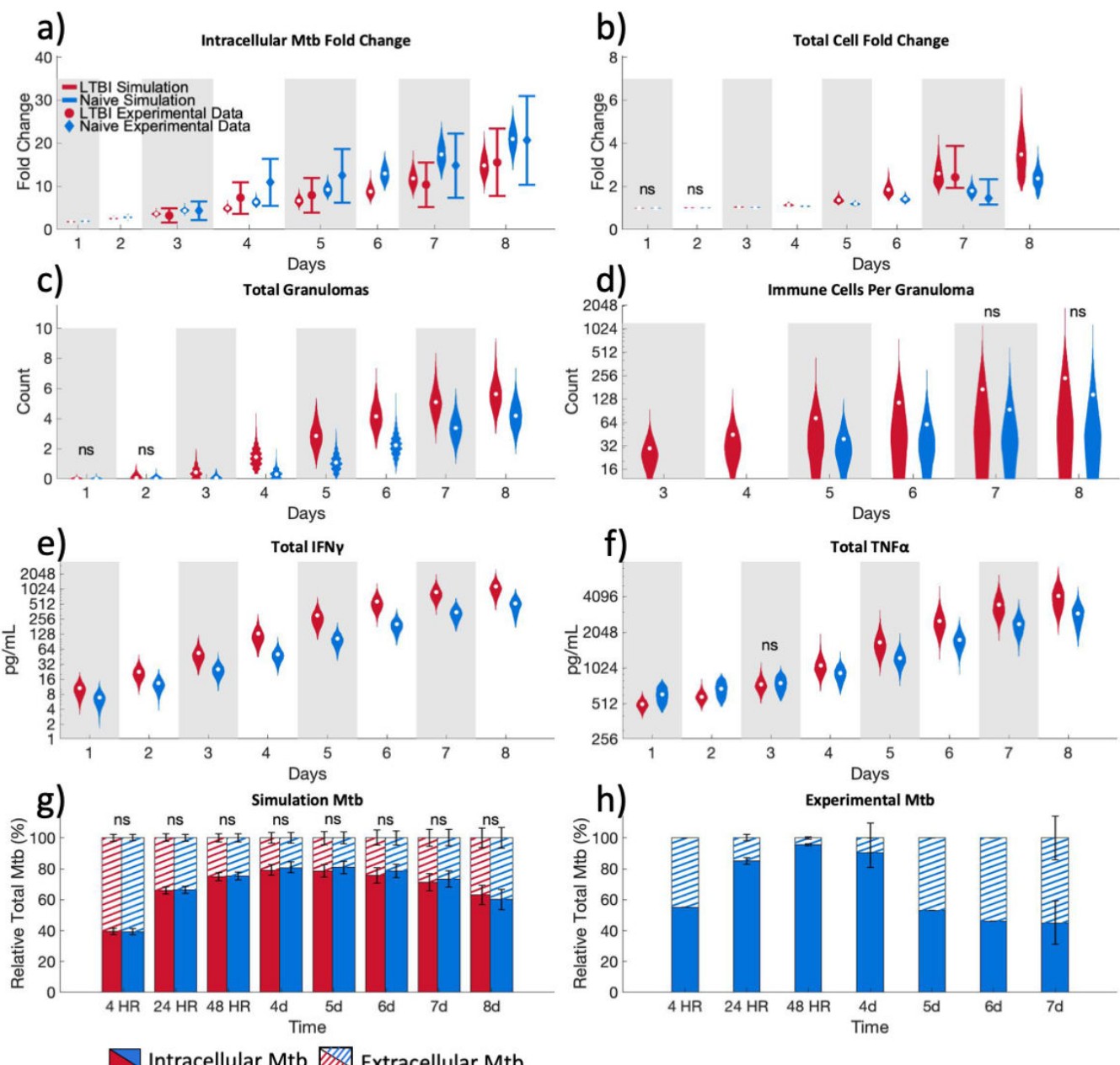

**FIG 2** Simulations mirror *in vitro* LTBI and naïve group experimental outcomes in calibration and validation data sets. Simulations compared to experimental data points for (a) CFU fold change and (b) total cell fold change as described in "Calibration." Mean experimental data obtained on days 3–5, 7–8 (a), and 7 (b) are from Guirado et al. (26). LTBI is shown in red circles, and mean experimental data for naïve are shown in blue diamonds on panels a and b. Error bars on the experimental data indicate: 0.5×–1.5× average experimental data point for panel a and 0.6×–1.8× for panel b. Violin plots visualize the distribution of the average of three replicates for the parameter sets that passed the calibration criteria with the white dots representing the mean. Qualitative validation is shown in panel c the average total number of granulomas per simulation, (d) the total number of immune cells (CD4 +T cells and macrophages) per granuloma, inflammatory cytokines (e) IFNγ, and (f) TNFα. Experimental validation compares (g) the relative percentage of total *Mtb* in LTBI and naïve simulations for intracellular and extracellular *Mtb* to (h) the relative percentage of intracellular and extracellular *Mtb* CFUs from experimental data. Experimental validation data in panel h were done with naïve cells only and are mean ± SEM of *n* = 1 (4 hours, 5 days, and 6 days) or 2 (24 hours, 48 hours, and 7 days). (a, b, c, e, and f) Simulation sample sizes are LTBI: *n* = 2,223; naïve: *n* = 2,099. (d) Simulation sample sizes are LTBI *n* = 31,312; Naïve *n* = 20,364. (g) Bars represent average of all passed parameter sets ± SD. (d through f) *Y*-axis is shown in base-2 log scale. Only non-significance is shown (ns); otherwise, the comparison between LTBI and naïve groups is meaningfully significant with a scaled A-measure of stochastic superiority above 0.71. See "Statistical analysis" for details on what we define as meaningfully significant.

experimental data, our simulations show lower bacterial fold changes and higher host cell fold changes in LTBI compared to naïve simulations.

As qualitative validation, our ABM predicts more (Fig. 2c) and larger (Fig. 2d) granuloma structures in LTBI simulations compared to naïve. The latter prediction agrees with experimental data that showed larger granuloma-like structures in LTBI cultures (26). Our simulations also predict higher IFNγ and TNFα concentrations (Fig. 2e and f) in LTBI compared to naïve simulations, in qualitative alignment with experimental data (26) that was not used in calibration. Finally, experimental trends in the percentage of intracellular and extracellular bacteria in cultures (see "Experimental validation") are consistent with our simulation predictions (Fig. 2g and h). Both experiments and simulations show a progressive increase in the percentage of intracellular bacteria through the first 4 days, with declining intracellular percentages thereafter, likely due to granulomas disaggregating and macrophages bursting.

These calibration and validation results confirm that our ABM can replicate and predict key experimental differences between LTBI and naïve data sets. We next used the LTBI and naïve simulations to further investigate which cellular mechanisms are predicted to contribute to these differences between LTBI and naïve responses to *Mtb*.

## LTBI simulations show that enhanced bacterial killing is a result of early CD4+ T cell interaction with infected macrophages which leads to CD4+ T cell and macrophage activation

In our simulation, bacterial growth inhibition is achieved through bacterial killing by macrophages (see *"Mycobacterium tuberculosis"* for more details). Therefore, we investigated whether the observed differences in the bacterial fold change between LTBI and naïve simulations are due to the increased killing of bacteria by macrophages in the LTBI simulations. To account for different bacterial loads between LTBI and naïve simulations (Fig. S2a through c at https://doi.org/10.5281/zenodo.13844841), we normalized the bacteria killed per day to the sum of the total bacteria in the simulation plus what was killed that day. This comparison showed that in LTBI simulations, a larger percentage of bacteria are killed, compared to naïve simulations (Fig. 3a). LTBI simulations kill more bacteria with fewer infected macrophages compared to naïve simulations until day 7 post-infection (Fig. S3a and b at https://doi.org/10.5281/zenodo.13844841).

Both resting and activated macrophages can kill bacteria in our simulations. Our results show that the contribution of activated macrophages to bacterial killing increases earlier in LTBI simulations compared to naïve simulations (Fig. 3b). Activated macrophages also account for a higher percentage of *Mtb* deaths per day between days 2 and 6 post-infection in LTBI simulations compared to naïve simulations. The earlier increase in the percentage of bacteria killed by activated macrophages in LTBI simulations also coincides with the increase in numbers of both total activated macrophages (Fig. S4a at https://doi.org/10.5281/zenodo.13844841) as well as activated infected macrophages (Fig. 3c). Infected macrophage activation in the naïve simulations lag by approximately 2 days behind the LTBI simulations. The activation of infected macrophages in LTBI and naïve simulations also coincides with the timing of the first granuloma formation (Fig. 2c). These results indicate that early macrophage activation in LTBI simulations is limiting bacterial growth.

Since there is a significant difference in the number of activated macrophages between LTBI and naïve simulations, we determined which of the two macrophage activation pathways in our simulations, NF-κB or STAT1 (see "Activation status"), is the limiting factor in total macrophage activation. The limiting factor in macrophage activation is the STAT1 pathway (Fig. S4b and c at https://doi.org/10.5281/zenodo.13844841). The STAT1 pathway is activated by the local concentration of IFNγ which is only produced in the simulation by activated CD4+ T cells. This prompted us to evaluate the number of activated TB-specific CD4+ T cells, which were significantly higher in LTBI simulations compared to naïve simulations throughout the entire simulation (Fig. 3d). This suggests that a small difference in activated CD4+ T cells in the first few days following infection has a large impact on the availability of IFNγ (both the total amount [Fig. 2e] and number of grid squares above the threshold for STAT1

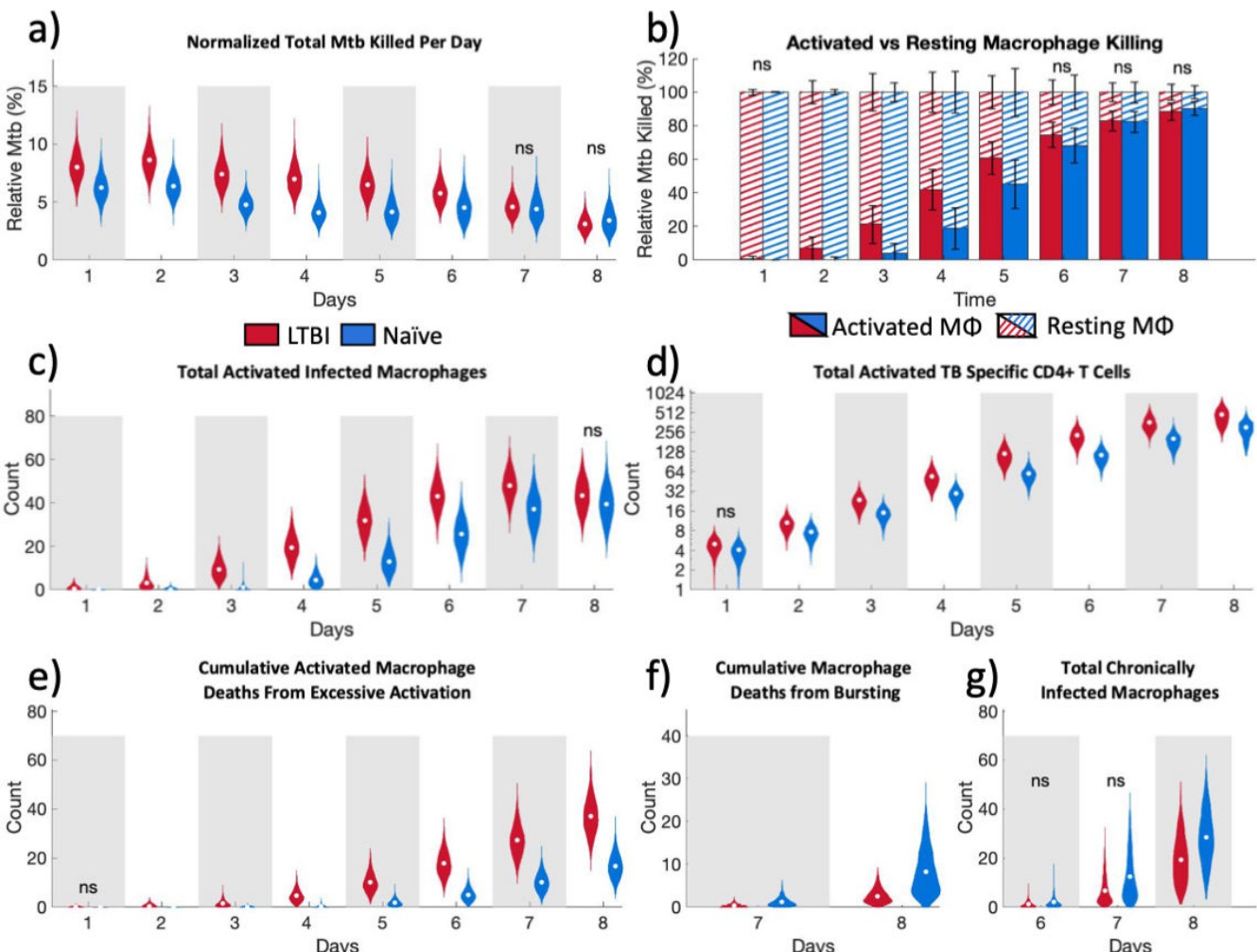

**FIG 3** Aggregate data comparisons between LTBI and naïve simulations show LTBI simulations kill more bacteria as a result of early CD4+ T cell interaction with infected macrophages leading to CD4+ T cell and macrophage activation. (a) Bacteria killed per day as a fraction of all the bacteria (live and dead) at that timepoint. (b) Percentage of total *Mtb* killed per day contributed by activated and resting macrophages. (c) Total number of activated infected macrophages. (d) Total activated TB-specific CD4+ T cells. (e) Cumulative activated macrophage deaths from excessive activation. (f) Cumulative total macrophage death due to bursting days 7–8. (g) Chronically infected macrophages days 6–8. (a, c through g) Results are of averaged outputs per parameter set that passed calibration. LTBI: $n$ = 2,223; naïve: $n$ = 2,099. (b) Bars represent average of all passed parameter sets ± SD. (d) *Y*-axis is shown in base-2 log scale. Only non-significance is shown (ns); otherwise, the comparison between LTBI and naïve groups is meaningfully significant with a scaled A-measure of stochastic superiority above 0.71. See Statistical analysis for details on what we define as meaningfully significant.

macrophage activation [Fig. S5 at https://doi.org/10.5281/zenodo.13844841]) which is the limiting factor to complete macrophage activation.

Bacterial killing declines in both LTBI and naïve simulations over time, albeit due to different reasons. In LTBI simulations, between days 7 and 8 post-infection, the number of activated infected macrophages decreases (Fig. 3c). This decrease in activated infected macrophages coincides with large numbers of macrophage deaths. In the simulation, there are two ways for macrophages to die: (i) from excessive activation and (ii) reaching a maximum threshold of intracellular bacteria that causes the macrophage to burst. LTBI simulations have increased cumulative activated macrophage death from excessive activation compared with naïve simulations (Fig. 3e) which coincides with the decrease in activated infected macrophages in LTBI simulations between days 7 and 8. In contrast, in naïve simulations, there is only a slight increase in the total activated infected macrophages at day 7–8 post-infection (Fig. 3c). Therefore, the decrease in bacterial killing is not associated with less activated infected macrophages but rather

is associated with increased cumulative total macrophage death due to bursting (Fig. 3f) and more chronically infected macrophages (Fig. 3g) at these later timepoints. Thus, bacterial killing is declining in LTBI simulations due to the loss of macrophages through excessive activation and in naïve simulations due to bursting and chronically infected macrophages that cannot fight infection. This shows that while macrophages are becoming activated *en masse* at later time points in the naïve simulations, and thus becoming better at killing, this occurs too late to suppress bacterial numbers below the bursting thresholds.

Taken together, this suggests that earlier contact between CD4+ T cells and infected macrophages in LTBI simulations enables earlier activation of infected macrophages and that, in turn, increases the ability of infected macrophages to kill intracellular bacteria and prevent bacterial growth. The downside to early macrophage activation in LTBI simulations is early macrophage death from excessive activation that impedes bacterial killing through the loss of the macrophages at later time points in this *in vitro* context. In contrast, macrophage activation being limited to later time points in naïve simulations means that effective activation occurs too late after the intracellular bacterial limit is reached and the macrophage bursts, also hampering bacterial killing. A key observation from this aggregate level of data is that the increase in macrophage activation in both LTBI and naïve simulations (Fig. 3c) coincides with the formation of granulomas (Fig. 2c). Therefore, we next quantified the influence of granuloma structure on macrophage activation.

## LTBI granuloma-like structures support quicker macrophage activation

We analyzed granuloma structure-specific metrics to determine how individual granuloma-like structures contribute to host responses in LTBI and naïve simulations.

Similar to the aggregate data (see "LTBI simulations show that enhanced bacterial killing is a result of early CD4 T cell interaction with infected macrophages which leads to CD4+ T cell and macrophage activation"), individual granuloma-like structures in the LTBI simulations have fewer *Mtb* than granuloma-like structures in naïve simulations (Fig. 4a). In both LTBI and naïve simulations, the majority of the bacteria in granuloma-like structures is intracellular (LTBI ≥ 75%; naïve > 70%), but the fraction of intracellular bacteria decreases between days 5 and 8 (Fig. 4b), similar to the aggregate metrics (Fig. 2g). However, unlike at the aggregate level, the number of activated infected macrophages per granuloma structure is lower in LTBI granuloma-like structures compared to the naïve simulations (Fig. 4c), although not meaningfully significant. This lower number of activated infected macrophages in LTBI granuloma-like structures is counterintuitively associated with more activated CD4+ T cells per granuloma-like structure compared to naïve simulations (Fig. 4d).

To characterize the role of activated CD4+ T cells in the granuloma-like structures, we assessed the distribution of immune cells per individual granuloma-like structure. CD4+ T cells make up the majority of the total immune cells per granuloma-like structure in both LTBI and naïve simulations, with LTBI simulations having a higher percentage than naïve simulations (Fig. 4e). TB-specific CD4+ T cells become the majority of these granuloma CD4+ T cells around day 5 for both LTBI and naïve, but with LTBI simulations having a higher percentage than naïve simulations (Fig. 4f). However, as the TB-specific CD4+ T cells proliferate and their percentage of the granuloma structure immune cells grows, the activated fraction of total CD4+ T cells remains relatively constant in both LTBI and naïve simulations (Fig. 4g) and is not meaningfully significant between LTBI and naïve simulations. This is also reflected in the decline in the percentage of activated, relative to total TB-specific CD4+ T cells in granuloma-like structures (Fig. S6 at https://doi.org/10.5281/zenodo.13844841). Thus, even though the LTBI simulations have a higher number of activated TB-specific CD4+ T cells compared to naïve simulations, and this number is increasing over time (Fig. 4d), the fraction of total TB-specific CD4+ T cells that are activated is decreasing over time. This suggests that the expansion of the TB-specific CD4+ T cell population does not effectively remain activated.

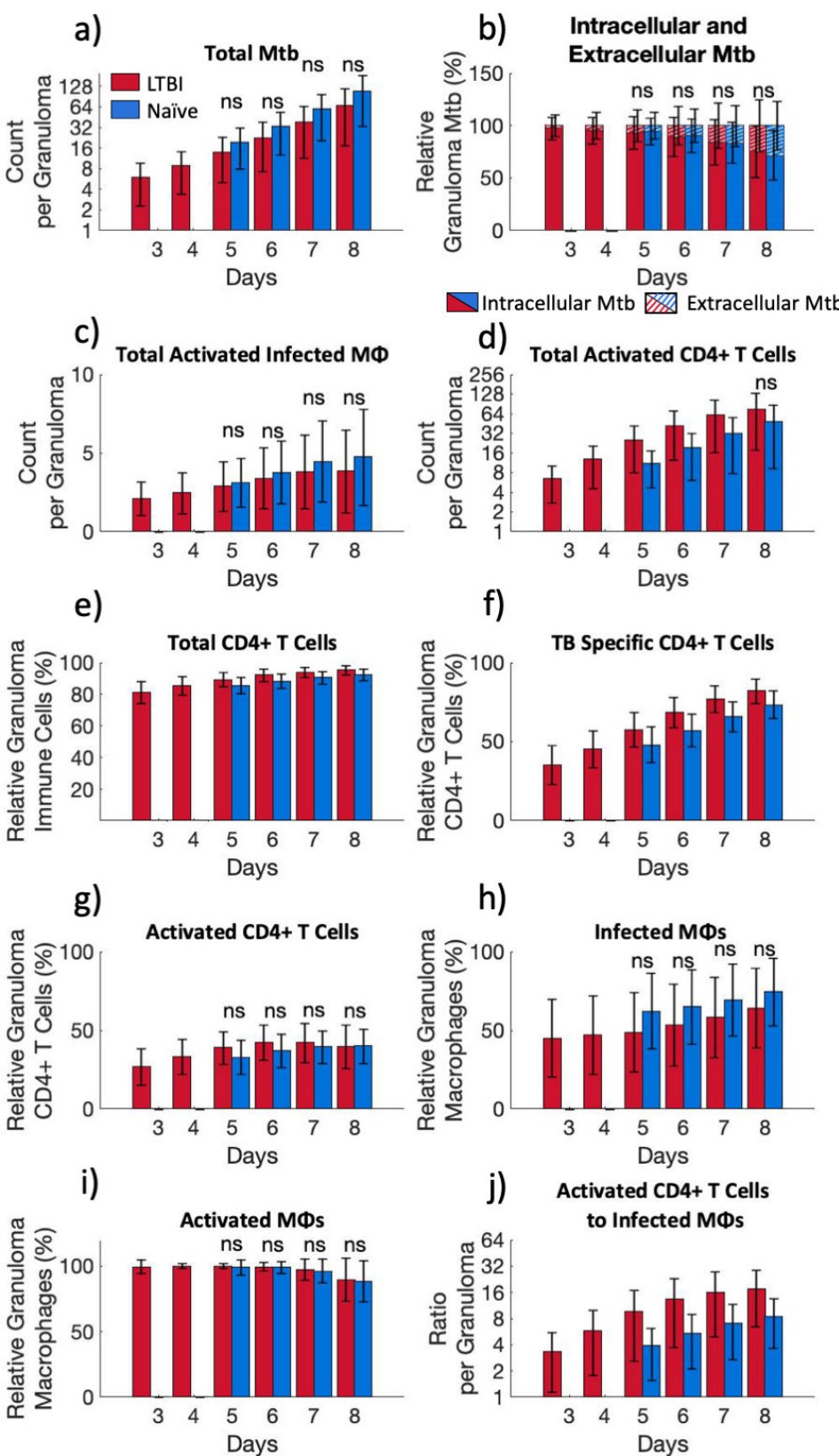

FIG 4  Individual granuloma-like structures between LTBI and naïve simulations differ. Per individual granulomas: (a) total *Mtb*, (b) fraction of intracellular and extracellular *Mtb*, (c) activated infected macrophages, and (d) activated CD4+ T cells. (e) Total CD4+ T cells as a fraction of all immune cells in an individual granuloma. (f) TB-specific CD4+ T cells as a fraction

Fig 4 (Continued)

of the granuloma CD4+ T cell population. (g) Activated TB-specific CD4+ T cells as a fraction of the granuloma CD4+ T cell population. (h) Infected macrophages as a fraction of the granuloma macrophage population. (i) Activated macrophages as a fraction of the granuloma macrophage population. (j) The ratio of activated CD4+ T cells to infected macrophages per granuloma. Bar graphs show the average and SD of all granulomas (LTBI $n$ = 31,312; Naïve $n$ = 20,364) in passed runs. Only non-significance is shown (ns); otherwise, the comparison between LTBI and naïve groups is meaningfully significant with a scaled A-measure of stochastic superiority above 0.71. See "Statistical analysis" for details on what we define as meaningfully significant. (a, d, and j) $Y$-axis is shown in base-2 log scale.

To assess the impact of TB-specific CD4+ T cell activation on macrophage activation within granuloma-like structures, we further characterized infected macrophage activation. LTBI simulations have a smaller percentage of macrophages that are infected per granuloma compared to naïve simulations throughout the infection (Fig. 4h), which could explain the lower numbers of activated infected macrophages (Fig. 4c). Indeed, regardless of infection status, nearly 100% of macrophages in the granuloma are activated in both LTBI and naïve simulations (Fig. 4i). This is supported by high ratios of CD4+ T cells to infected macrophages in LTBI compared to naïve granuloma structures (Fig. 4j). Therefore, macrophage activation does not appear to be the limiting factor in bacterial killing within granuloma-like structures.

As a whole, these results suggest that there are spatial patterns that explain the decreasing fraction of activated TB-specific CD4+ T cells and how this decreasing fraction of activated T cells still manages to activate nearly all macrophages in the granuloma structures.

To look for spatial patterns, we evaluated representative snapshots of granuloma structures (Fig. 5a through f). The structural organization of the granuloma-like structures shows that CD4+ T cell activation is centered in the granuloma structures (teal cells in Fig. 5d through f, relative to the position of orange cells in Fig. 5a through c) with unactivated TB-specific CD4+ T cells around the periphery (light blue cells in Fig. 5d through f). Furthermore, the results indicate that there are short distances between activated infected macrophages and activated CD4+ T cells (cyan to hot pink cells in Fig. 5d through f). From these images, other observations include LTBI granuloma structures are more closely packed compared to the naïve granuloma structures, and the majority of all the activated CD4+ T cells are within granuloma structures for both LTBI and naïve groups.

Since the activation of macrophages and CD4+ T cells is centered in the granuloma, we analyzed the concentration gradients of TNFα (Fig. 5g and h) and IFNγ (see Fig. S7 at https://doi.org/10.5281/zenodo.13844841) to see if these cytokines are also localized to the granuloma. Results indicate that the TNFα concentration is much higher in the granuloma structures compared to outside the structure (Fig. 5g and h). Additionally, the IFNγ threshold for STAT1 activation (white contour lines in Fig. 5g and h) bounds the granuloma structure. This suggests that the IFNγ signal to activate macrophages is also localized to the granuloma structure. Thus, T cells at the periphery of granulomas would be within TNFα concentrations that are high enough to prevent movement away from the granuloma structures but are too far away from infected macrophages to maintain their activation status. This observation is true for both LTBI and naïve simulations (Fig. 5).

In summary, the structure of the granuloma is important for creating an environment that promotes co-localization and activation of both CD4+ T cells and infected macrophages, despite progressive loss of activation of CD4+ T cells in the periphery of the granuloma-like structures.

## TNFα and IFNγ related parameters along with bacterial killing probability drive increased bacterial killing in LTBI simulations

To identify the simulation-predicted mechanisms that underlie the differences in bacterial control, proliferative activity, and granuloma structure formation between

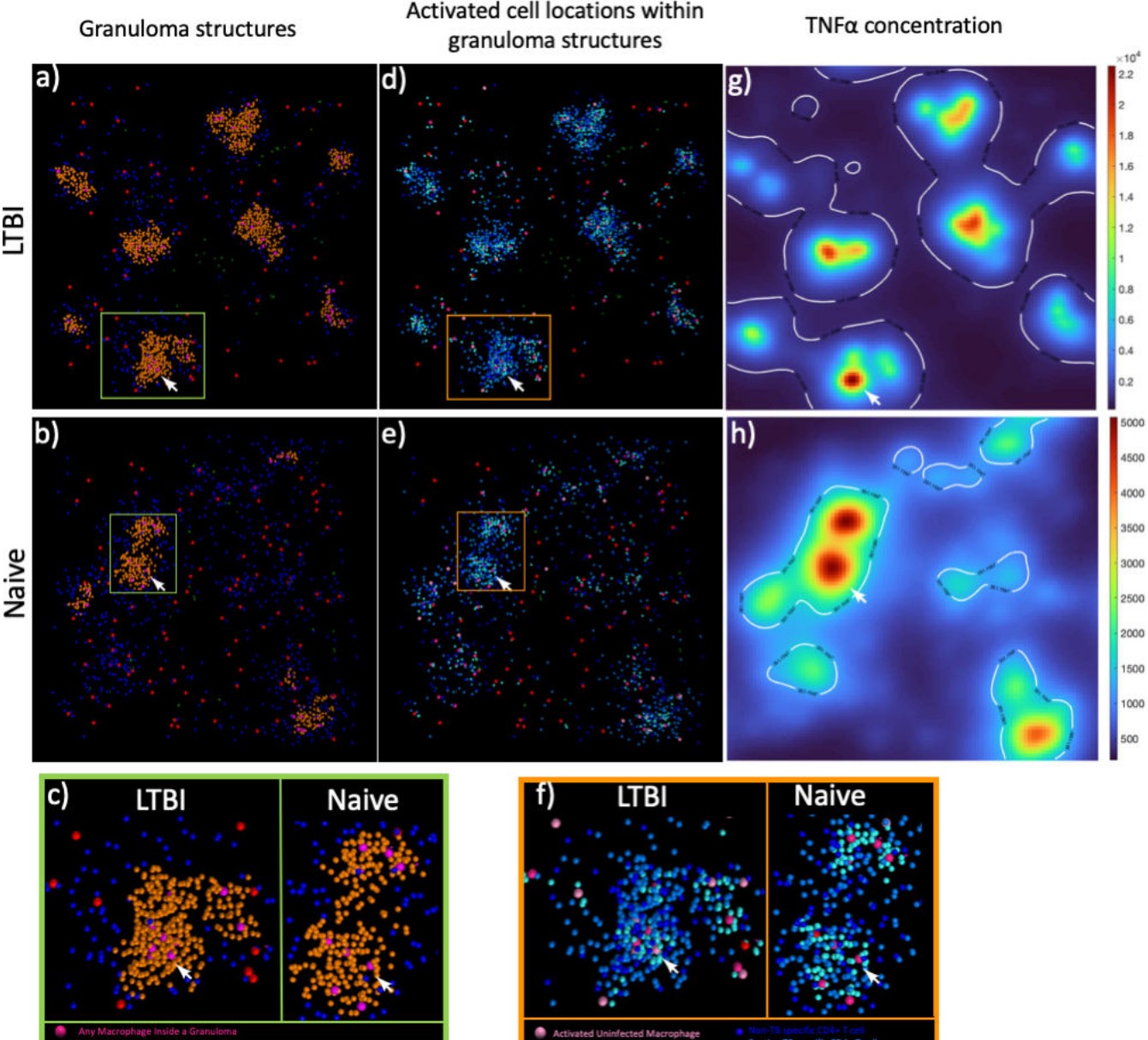

**FIG 5** Granuloma-like structures are different between LTBI and naïve simulations. (a and b) Granuloma-like structures for LTBI (a) and naïve (b) that highlight macrophage and CD4+ T cells inside (orange and hot pink) vs outside (blue and red) granuloma structures. (c) Zoom in on granuloma structure in green box in panels a and b. Using panels a and b as an outline for cells in granuloma structures, panels d and e are the corresponding image with all cell states to distinguish the location of infected macrophages, activated macrophages, and activated TB-specific CD4+ T cells in granuloma structures for LTBI (d) and naïve (e). (f) Zoom in on granuloma structure in orange box in panels d and e. (g and h) corresponding TNFα concentration gradient for (g) LTBI and (h) naïve simulations in panels d and e. The white line indicates the threshold IFNγ concentration for STAT1 activation in macrophages. All images show results for day 8 post-infection. White arrow denotes the same location within panels a through h.

LTBI and naïve experimental data, we assessed parameter differences between the calibrated LTBI and naïve simulations. As discussed in "Statistical analysis" above, we applied the A-measure of stochastic superiority to compare parameter distributions between LTBI and naïve simulations. Figure 6 shows the normalized parameter ranges for calibrated LTBI and naïve parameter sets for all parameters with a point estimate of the scaled A-measure of stochastic superiority greater than 0.71. Figure S8 at https://doi.org/10.5281/zenodo.13844841 shows the normalized parameter ranges for all 37 varied parameters, and Fig. S9 at https://doi.org/10.5281/zenodo.13844841 shows the

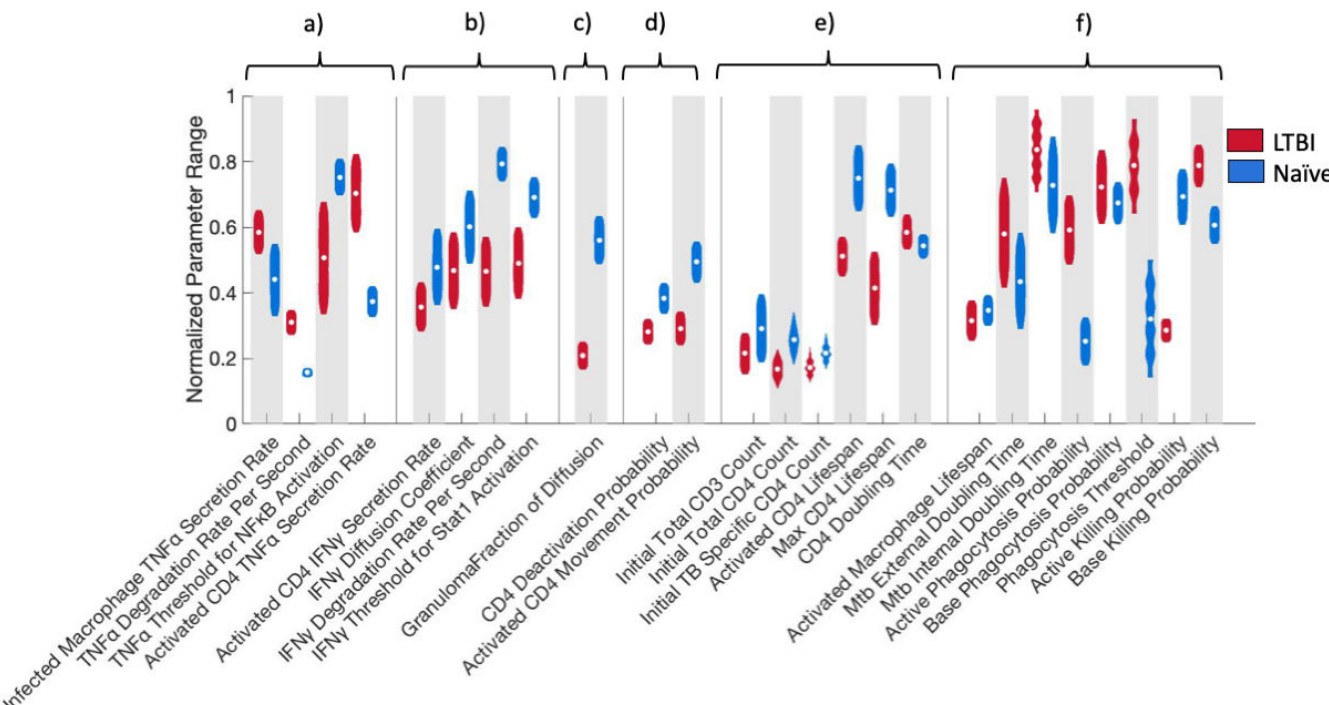

**FIG 6** Key parameter differences between LTBI and naïve-simulations that may drive bacterial killing, proliferative activity, and granuloma formation in LTBI simulations include higher TNFα secreted from infected macrophages, lower IFNγ threshold for STAT1 activation, lower probability of CD4 deactivation, and higher baseline killing probabilities. Normalized parameter ranges for parameters above 0.71 for the scaled A-measure of stochastic superiority for (a) TNFα parameters, (b) IFNγ parameters, (c) granuloma structure parameter, (d) CD4+ T cell parameters, (e) initialization parameters, and (f) macrophage parameters. Violin plots visualize the distribution of the average of three replicates for the parameter sets that passed the calibration criteria with the white dots representing the mean. LTBI $n$ = 2,223, naïve $n$ = 2,099. See https://doi.org/10.5281/zenodo.13844841 for details on calculation and "Statistical analysis" for details on what we define as meaningfully significant.

parameter point estimate of the scaled A-measure of stochastic superiority for all varied parameters. In the following sections, we will discuss groups of parameters and their potential influence on (i) initial contact between infected macrophages and CD4+ T cells and (ii) the organization and function of granuloma-like structures such as strong chemokine signals within the granuloma structures and short distances between activated infected macrophages and activated CD4+ T cells (Fig. 4 and 5).

### TNFα-related parameters may attract CD4+ T cells to infected macrophages better in LTBI simulations

Infected macrophages in LTBI simulations secrete more TNFα than in naïve simulations and TNFα degrades more quickly in LTBI simulations compared to naïve simulations (Fig. 6a). This combination of parameter differences would result in a steeper TNFα gradient in LTBI simulations compared to naïve simulations (Fig. S10 at https://doi.org/10.5281/zenodo.13844841; based on equation B3.11 in reference 90). This steeper gradient could aid TB-specific CD4+ T cells to find infected macrophages more quickly in LTBI simulations. LTBI simulations also have a lower TNFα threshold to activate the NF-κB pathway in macrophages (Fig. 6a). Although the NF-κB pathway is not the limiting factor to macrophage activation, this threshold difference could contribute to the lack of early macrophage NF-κB activation in the naive simulations (Fig. S4b). These TNFα related mechanisms therefore combine to predict that LTBI simulations are better able to colocalize CD4+ T cells near infected macrophages while activating NF-κB in macrophages in the same area, as compared to naïve simulations. This TNFα-driven advantage in LTBI simulations is further amplified once CD4+ T cells get activated since

activated CD4+ T cells secrete more TNFα (Fig. 6a) in LTBI simulations compared to naive simulations.

### IFNγ-related parameters support faster infected macrophage activation in LTBI simulations after initial contact between infected macrophage and activated CD4+ T cells

In contrast to TNFα, LTBI simulations have lower IFNγ secretion from activated CD4+ T cells compared to naïve simulations (Fig. 6b). However, IFNγ diffuses and degrades more slowly in LTBI simulations (Fig. 6b) meaning that the lower amount of IFNγ produced by a single CD4+ T cell in LTBI simulations does not quickly dissipate from the secreting cell. IFNγ is necessary in our simulations to activate the STAT1 pathway in macrophages. The IFNγ threshold for macrophage STAT1 activation is also lower in LTBI simulations (Fig. 6b) indicating that even though CD4+ T cells secrete less IFNγ, less IFNγ is necessary to activate macrophages. In addition, since there are more activated CD4+ T cells secreting IFNγ beginning quickly after initial contact between infected macrophages and CD4+ T cells in LTBI simulations, there is more IFNγ available in the simulation in total (Fig. 6e). Our model therefore suggests that these IFNγ parameters are the driving forces behind more activated macrophages and activated CD4+ T cells in LTBI simulations. These predicted parameter impacts are supported by our sensitivity analysis. Partial rank correlation coefficients (Fig. S11 at https://doi.org/10.5281/zenodo.13844841) revealed that the IFNγ diffusion coefficient, IFNγ degradation rate, and IFNγ threshold for STAT1 activation were negatively correlated with the number of activated macrophages, while CD4+ T cell IFNγ secretion was positively correlated with the number of activated macrophages.

### Diffusion within a granuloma is slowed in LTBI simulations

The fraction of the diffusion coefficient (same for TNFα and IFNγ) to diffuse through a granuloma is lower in LTBI simulations meaning that the cytokines diffuse more slowly through granuloma-like structures in LTBI simulations compared to naïve simulations (Fig. 6c). This can be seen in Fig. 6 and Fig. S7 at https://doi.org/10.5281/zenodo.13844841 where the cytokine concentrations of both TNFα and IFNγ are more centrally located and higher in LTBI simulations compared to naive simulations. Indeed, our simulations of LTBI granuloma structures seem to be more densely packed together compared to naïve simulations. This paired with the lower TNFα degradation supports our theory that the granuloma structure provides a strong localized chemokine signal to retain immune cells in the granuloma structure. However, this diffusion fraction was not correlated with any outputs in our sensitivity analysis.

### CD4+ T cell parameters help to sustain activated CD4+ T cell status better in LTBI simulations

Activated CD4+ T cells in LTBI simulations have a lower probability of deactivating and a lower probability of moving compared to naïve simulations (Fig. 6d). This again contributes to a higher chance that in the LTBI simulations, the activated CD4+ T cells will stay close to infected macrophages (i) sustaining the IFNγ needed to maintain STAT1 activation and therefore total macrophage activation leading to bacterial killing and (ii) forming granulomas more quickly as the CD4+ T cells start to proliferate and have a small probability of moving away from the granuloma due to lower movement and a high TNFα signal creating a strong gradient to the granuloma. These relationships are also reflected in our sensitivity analysis that showed that activated CD4+ T cells are negatively correlated with CD4 deactivation probability and activated CD4+ T cell movement probability (Fig. S12 at https://doi.org/10.5281/zenodo.13844841).

## T cell initial conditions might reflect different T cell subsets, but starting numbers are less important than cell responsiveness to stimuli

Additional parameters related to CD4+ T cells that are stochastically different between LTBI and naïve simulations include the starting number of CD3+ T cells (only used to determine number of CD4+ T cells), CD4+ T cells, starting number of TB-specific CD4+ T cells, CD4+ T cell-activated lifespan, CD4+ T cell doubling time, and CD4+ T cell max lifespan (Fig. 6e). These are all expected to influence the quantity of CD4+ T cells but, interestingly, are all lower in the LTBI simulations. Since LTBI simulations start with fewer CD4+ T cells and end with far more CD4+ T cells than naïve simulations, these parameter differences might be more reflective of different phenotypes of TB-specific CD4+ T cells which is not explicitly accounted for in the current ABM.

## Macrophage base killing and phagocytosis parameters directly impact bacterial population

The parameters described so far indirectly influence bacterial killing through macrophage activation. In our simulation, macrophages are the only cell type that directly influences the bacterial population through phagocytosis and killing. The macrophage-activated lifespan and *Mtb* intracellular and extracellular bacterial doubling times are considered stochastically different; however, the macrophage maximum activated lifespan is longer (10+ days) than the simulated time, so it is not considered to be important for our conclusions here. Interestingly, the *Mtb* external and internal growth rates were slower for LTBI simulations compared to naïve (Fig. S8 at https://doi.org/10.5281/zenodo.13844841). To confirm that the growth rates did not affect our conclusions, we matched simulations between LTBI and naïve with the same growth parameters and determined that our conclusions remain unchanged (see Table S3 at https://doi.org/10.5281/zenodo.13844841). LTBI simulations have a higher base and activated macrophage phagocytosis probability and a higher phagocytosis threshold (Fig. 6f). Indeed, there are more phagocytosis events by activated macrophages in LTBI simulations, both at the aggregate level and in the granuloma (Fig. S13 at https://doi.org/10.5281/zenodo.13844841). However, since this does not result in more intracellular bacteria or more infected macrophages (Fig. S2b and S3a at https://doi.org/10.5281/zenodo.13844841) in the LTBI simulations, it may only be helpful in the granuloma when macrophages die of excessive activation, and the surviving macrophages can still phagocytose bacteria, whereas macrophages in naïve simulations may not due, to this lower threshold. LTBI simulations have a higher base killing probability, but counterintuitively, a lower active killing probability compared to naïve simulations (Fig. 6f). This difference does not seem to affect the bacterial load until late in the infection as naïve simulations have significantly fewer activated macrophages throughout the course of infection (Fig. 3c; Fig. S4a at https://doi.org/10.5281/zenodo.13844841). The simulation-predicted order of killing probabilities is as follows from worst to best: naïve base killing probability < LTBI base killing probability < LTBI activated killing probability < naïve active killing probability. Throughout most of the simulation, the macrophages in LTBI simulations kill at two different probabilities, both better than the macrophages in naïve simulations killing at their baseline probability and with fewer total infected macrophages (Fig. S3a at https://doi.org/10.5281/zenodo.13844841). This highlights the importance of bacterial killing probabilities and activation status. It is only when the number of activated macrophages in naïve simulations increases around day 6 (Fig. 3c) that they begin to utilize the highest killing probability in the system and kill bacteria more efficiently and pass the LTBI for most bacteria killed (Fig. S3b at https://doi.org/10.5281/zenodo.13844841). However, as mentioned above, this might come too late, after the bacterial population left to kill is too large. Our sensitivity analysis confirms that the base killing probability is positively correlated with total *Mtb* killed and negatively correlated with total *Mtb* (Fig. S14 at https://doi.org/10.5281/zenodo.13844841).

## Summary

Taken together, these stochastically different parameters paint the picture that differences between LTBI and naïve simulation outcomes rely heavily on macrophage killing probabilities, TNFα and IFNγ parameters, specifically infected macrophage TNFα secretion, and IFNγ threshold for STAT1 macrophage activation. Secretion, diffusion, and degradation-related parameters for both TNFα and IFNγ seem to be the driving forces behind early initial contact between infected macrophages and CD4+ T cells and strong cytokine signals within granulomas that lead to more activated macrophages, more activated CD4+ T cells, quicker granuloma formation, and more bacterial control in LTBI simulations. As the CD4+ T cells proliferate, they are less likely to navigate away from the granuloma-like structure due to their lower movement probability and the compounding amount of TNFα (from activated CD4+ T cells and activated macrophages) being produced at these sites, thus sustaining the granuloma cellular population.

## DISCUSSION

We have developed an ABM to investigate potential mechanistic differences in early immune responses between cells from LTBI vs naïve individuals. Our model emulates an *in vitro* experimental system of early infection dynamics and granuloma formation. We have calibrated our model to experimentally measured bacterial fold change, cellular proliferation, and granuloma structure formation. We performed this calibration separately for LTBI and naïve data sets (26). We have qualitatively validated our ABM by predicting additional features of the *in vitro* experimental system including higher inflammatory cytokine levels of IFNγ and TNFα and larger-sized granuloma structures in the LTBI simulations compared to naïve simulations.

We identify several LTBI-associated mechanisms that contribute to fewer bacteria, more proliferative activity, and faster and larger granuloma-like structures including: (i) earlier initial contact between T cells and infected macrophages relative to the exposure event, (ii) short distances between activated infected macrophages and activated T cells within the granuloma, and (iii) effective chemokine gradients directing recruited cells toward intracellular bacteria within infected macrophages. In naïve simulations, CD4+ T cells take approximately 2 days longer to reach *Mtb*-infected macrophages, therefore allowing a window for bacterial load to increase. Thus, our results underscore that the initial interactions between macrophages and CD4+ T cells (in the first 3 days) are important for jump-starting the antimycobacterial processes.

TNFα plays an important role in the recruitment and activation of T cells and macrophages and in maintaining the granuloma structure (91–96). TNFα-deficient mice are extremely susceptible to TB with high mortality rates and do not form granulomas (97, 98). TNFα neutralizing agents in animal models (97, 99, 100) following bacillus Calmette-Guérin (BCG) vaccination (101) and in LTBI individuals (102–107) have shown worsening disease, dissolution of granulomas, *Mtb* dissemination, and reactivation of TB. Our simulations are in alignment with this experimental evidence. We predict that earlier activation of CD4+ T cells in LTBI simulations leads to earlier activation of infected macrophages resulting in better bacterial killing in LTBI simulations compared to naïve simulations. Based on simulation parameter estimates, these differences are due to a higher TNFα secretion rate from infected macrophages and a steeper TNFα gradient that attracts and localizes CD4+ T cells and thus activates the CD4+ T cells in LTBI. In return, the IFNγ secreted from these newly activated CD4+ T cells activates the limiting STAT1 pathway for complete macrophage activation initiating an increase in the macrophage bacterial killing rate. Thus, we show how specific TNFα-associated mechanisms could contribute to different early responses in cells from LTBI vs naïve individuals.

IFNγ is crucial for host protection against *Mtb* (108–110). IFNγ-deficiency makes the host highly susceptible to *Mtb* infections and leads to poor outcomes. Similar to TNFα, IFNγ enhances the anti-mycobacterial ability of macrophages and plays an important role in granuloma development (109, 110). IFNγ produced by CD4+ and CD8+ T cells is

critical compared to other sources of IFNγ (111–113). Our model predicts that IFNγ-associated mechanisms also contribute to early immune response differences between cells from LTBI and naïve individuals. Macrophages in the LTBI simulations have a lower IFNγ threshold for activation, suggesting an increased sensitivity to IFNγ. Unexpectedly, CD4+ T cells secrete less IFNγ per cell in LTBI simulations compared to naïve. This is negated by the fact that more cells are secreting IFNγ in LTBI simulations, so there is more IFNγ in the total simulation volume. Similarly, in mice, lower per-cell secretion of IFNγ had higher survival compared to overproduction of IFNγ per-cell (112).

These emergent differences in TNFα and IFNγ parameters between LTBI and naïve simulations could be reflective of immune memory in LTBI cells, as was hypothesized for the corresponding *in vitro* infection (26). We did not explicitly incorporate immune memory as a mechanism in our simulations due to a lack of data. Nonetheless, the two types of immune memory that could be represented in our model are (i) *Mtb*-specific CD4+ T cell memory and (ii) innate immune memory (also known as trained immunity). First, *Mtb*-specific CD4+ T cells are necessary for control of *Mtb* infection (114–119). While our LTBI simulations counterintuitively start with fewer overall *Mtb*-specific CD4+ T cells (Fig. 6e), it is possible that T cell phenotypes are more important than T cell numbers. Indeed, while we do not explicitly account for memory T cells, *Mtb*-specific CD4+ T cell memory could be reflected in our model by the more profound proliferation of *Mtb*-specific CD4+ T cells in LTBI simulations compared to naïve simulations (Fig. 2b).

The second form of immune memory represented in our simulations is related to trained immunity. In addition to adaptive immune memory, trained immunity has emerged as another potential control strategy in TB (120–124). Trained immunity occurs when innate immune cells are epigenetically reprogrammed during exposure to direct infection, vaccines, pathogen-associated molecular patterns, or cytokines, thereby allowing for enhanced responses upon secondary infection (120–123, 125). Trained immunity studies have primarily focused on long-term changes in bone marrow hematopoietic stem cells that lead to functional changes in short-lived innate immune cells during TB infection (20, 126, 127), as well as the effects of the BCG vaccine on innate immune efficacy in controlling *Mtb* growth (128–132), but not LTBI. Trained immunity via prior exposure to viruses can alter the functions of tissue-resident macrophages (20, 133–136) and occur at sites of infection or inflammation (125, 137). The presence of T cells (20, 136, 138) is also a key driver of trained immunity development. Thus, it seems plausible that in a persistent infection like LTBI, T cells, and granuloma structures are an exemplary environment for macrophages to develop trained immunity. Such trained immunity is reflected in our model in parameters such as a lower threshold of IFNγ to activate macrophages in our LTBI simulations (Fig. 3d and 6). Trained immunity is also associated with the upregulation of TLRs in individuals with LTBI compared to uninfected controls (139). In our ABM, the higher baseline macrophage bacterial killing probability and higher phagocytosis probabilities in the LTBI simulations could reflect this upregulation of TLRs in LTBI (Fig. 6). Since fewer macrophages were infected in the LTBI simulations, but bacterial growth was controlled better, this could reflect trained immunity where the "quality" of the macrophages' antimicrobial functionality was enhanced. To further elucidate the roles of innate vs adaptive immune memory in LTBI responses, more investigation into the potential of persistent infection to induce trained immunity will be informative.

Our collective findings described above are also in agreement with both a contained TB infection (CMTB) mouse model (18, 20) and a concurrent TB infection NHP model (19), where prior infection with *Mtb* offers protection against reinfection. First, in agreement with our simulations, the CMTB mouse model shows that innate functions are enhanced, and specifically, alveolar macrophage (AM) activation has faster initiation and greater numbers driven by low levels of inflammation; this activation is driven by IFNγ that is likely secreted by *Mtb*-specific T cells; fewer AMs from CMTB mice were infected during reinfection (18, 20). Second, the NHP model indicates that NHPs with a primary infection had a lower bacterial burden in new granulomas and increased bacterial killing

after reinfection compared to primary infection; this difference is due to the presence of *Mtb*-specific T cells and activated innate immune responses in the uninvolved lung (19). These NHP results are in accordance with our predictions that innate immune cell activation via *Mtb*-specific T cell responses results in increased bacterial killing and lower bacterial burden in LTBI simulations compared to naïve simulations.

Consistent with our predictions, experimental studies have demonstrated that activated macrophages lie almost exclusively at the center of the granuloma (140), T cells are predominantly found at the periphery of granulomas (141–143), the cellular microenvironment of the granuloma plays a key role in shaping the host immune response (141, 143–147), and direct contact between infected macrophages and CD4+ T cells is necessary for effective T cell responses to *Mtb* infection (114). Our model predictions are also consistent with experimental findings that lymphocytes in the granuloma cuff were rarely activated (141). BCG granulomas in mouse liver showed that relatively few T cells secrete IFNγ in a spatially polarized fashion, most likely resulting in a concentrated localized delivery to few antigen-presenting cells (148). This is in line with our predictions that few T cells toward the interior of granulomas are in contact with infected macrophages and secreting IFNγ.

Thus, while this spatial architecture of granulomas is beneficial to activate infected macrophages and CD4+ T cells in the core of the granulomas, the CD4+ T cells at the periphery are maintained near the granuloma by the TNFα gradients but not activated. This finding agrees with BCG-infected mouse liver granulomas showing that CD4+ T cells frequently enter but do not exit granulomas, and TNFα from infected cells serves as an early source of chemoattractant that initiates granuloma formation (91). Other computational models have also shown that the spatial organization of the granuloma leads to crowding and restricts the movement of T cells primarily to the border of the granuloma (140). We conclude that the spatiotemporal features of the granuloma in the first few days contribute to bacterial control and long-term outcomes and support the hypothesis that activation occurs only in the center of structures.

As with all computational models, there are limitations to our ABM. First, all cell types present within granulomas and whole blood were not modeled due to a lack of data, unknown functions of some of these cells within granulomas, and the computational and parameterization complexity associated with a larger group of cells. As the ABM develops, more cell types, such as neutrophils, dendritic cells, and natural killer cells, can systematically be included to reflect the population of cells within granulomas as data in these *in vitro* systems become available. Second, bacterial subpopulations and metabolic adaptions due to nutrient availability may impact bacterial growth (26, 149). However, to reduce the complexity of the simulations, we opted to assume that all growth inhibition would be captured by the simplest method using macrophage phagocytosis and killing probabilities. More detailed simulations of the role of nutrient and carbon sources on *Mtb* metabolic pathways (149, 150) could be included in future models to further expand our predictions. Our simulation, and the experimental system on which it is based, represent a closed system that does not currently allow for cell supplementation over time. Thus, our predictions are inherently limited to this closed-system scope. Nonetheless, the above discussion highlights *in vivo* findings (mouse and NHP) that qualitatively and conceptually support our findings.

We have demonstrated that differences between LTBI and naïve host responses rely heavily upon the initial interactions between macrophages and CD4+ T cells, and that these early interactions are driven by a combination of innate, adaptive, and spatial differences. Our findings as well as others support the importance of understanding the nature and the timing of initial interactions between the innate and adaptive immune cells (30–32, 151). Our results will help inform how insights from LTBI might be used to identify long-term protective immune responses. This, in turn, could support the development of effective vaccines or host-directed therapies that exploit the early interactions between the innate and adaptive systems such as increasing macrophage

autophagy (42, 152, 153) or exogenous IFNγ treatment (153–155) to reduce bacterial load.

## ACKNOWLEDGMENTS

We thank Lev Gorenstein and the rest of the Rosen Center for Advanced Computing research staff for their assistance with batch computing. We would also like to acknowledge Catherine Weathered for her mentorship and her work setting up the foundations in Repast and SLURM for our lab.

This project was funded with support from the Indiana Clinical and Translational Sciences Institute which is funded in part by Award Number TL1TR002531 from the National Institutes of Health, National Center for Advancing Translational Sciences, Clinical and Translational Sciences Award to A.H. The content is solely the responsibility of the authors and does not necessarily represent the official views of the National Institutes of Health. This work was partially funded by a PhRMA Foundation Research Starter Grant to E.P. and L.S.S., National Institutes of Health awards to L.S.S. (AI136831, AI145539, 1P30AI168439), and a Texas Biomedical Research Institute Forum Award to L.S.S. and E.A. Simulations were run using the Extreme Science and Engineering Discovery Environment (XSEDE), which is supported by National Science Foundation grant number ACI-1548562; Anvil at Purdue and Expanse at UCSD were used through allocation TG-MDE220002. This research was done using services provided by the OSG Consortium (156–159), which is supported by the National Science Foundation awards #2030508 and #1836650.

## AUTHOR AFFILIATIONS

[1]Weldon School of Biomedical Engineering, Purdue University, West Lafayette, Indiana, USA
[2]Texas Biomedical Research Institute, San Antonio, Texas, USA
[3]Regenstrief Center for Healthcare Engineering, Purdue University, West Lafayette, Indiana, USA

## AUTHOR ORCIDs

Alexis Hoerter http://orcid.org/0000-0001-6667-1796
Elsje Pienaar http://orcid.org/0000-0002-5408-8795

## FUNDING

| Funder | Grant(s) | Author(s) |
| --- | --- | --- |
| Indiana Clinical and Translational Sciences Institute (CTSI) | TL1TR002531 | Alexis Hoerter |
| HHS \| National Institutes of Health (NIH) | AI136831, AI145539, 1P30AI168439 | Larry S. Schlesinger |
| Pharmaceutical Research and Manufacturers of America Foundation (PhRMAF) | | Larry S. Schlesinger |
| | | Elsje Pienaar |
| Texas Biomedical Research Institute (Texas Biomed) | | Eusondia Arnett |
| | | Larry S. Schlesinger |

## AUTHOR CONTRIBUTIONS

Alexis Hoerter, Conceptualization, Data curation, Formal analysis, Methodology, Software, Validation, Visualization, Writing – original draft | Alexa Petrucciani, Methodology, Software, Writing – review and editing | Jordan Bonifacio, Data curation | Eusondia Arnett, Data curation, Writing – review and editing | Larry S. Schlesinger, Funding acquisition, Writing – review and editing | Elsje Pienaar, Funding acquisition, Supervision, Writing – review and editing

## DATA AVAILABILITY

All output data from simulations, supplemental material, calibrated parameter sets, and code for processing and visualization (MATLAB and Python scripts) are archived on Zenodo (10.5281/zenodo.13844841). The Repast model and uncalibrated parameter sets to run the code can be found at https://github.itap.purdue.edu/ElsjePienaarGroup/LTBI-NaiveinvitroModel.

## ADDITIONAL FILES

The following material is available online.

Open Peer Review

**PEER REVIEW HISTORY (review-history.pdf).** An accounting of the reviewer comments and feedback.

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
