## [Reviewer comments · mSystems]

Timing matters in Macrophage / CD4+ T cell interactions: An agent-based model comparing *Mycobacterium tuberculosis* host-pathogen interactions between latently infected and naïve individuals

Alexis Hoerter, Alexa Petrucciani, Jordan Bonifacio, Eusondia Arnett, Larry Schlesinger, and Elsje Pienaar

Corresponding Author(s): Elsje Pienaar, Purdue University

Review Timeline:

Submission Date:	October 4, 2024
Editorial Decision:	December 13, 2024
Revision Received:	December 16, 2024
Accepted:	December 17, 2024

Editor: Andrew Bartko

Reviewer(s): Disclosure of reviewer identity is with reference to reviewer comments included in decision letter(s). The following individuals involved in review of your submission have agreed to reveal their identity: Evan Skowronski (Reviewer #2)

Transaction Report:

DOI: <https://doi.org/10.1128/msystems.01290-24>

Re: mSystems01290-24 (Timing matters in Macrophage / CD4+ T cell interactions: An agent-based model comparing *Mycobacterium tuberculosis* host-pathogen interactions between latently infected and naïve individuals)

Dear Dr. Elsje Pienaar:

Revision Guidelines

Sincerely,
Andrew Bartko
Editor
mSystems

Reviewer #1 (Comments for the Author):

All major and almost all minor concerns have been addressed.

One minor issue that remains is that the authors state that they have substituted t-tests in Fig 2, 3, and 4 for the A-measure of stochastic superiority, yet the figure legend for Figure 3 still refers to t-test results.

Reviewer #2 (Comments for the Author):

The authors have responded in a thoughtful and thorough manner to the extensive comments contributed by Reviewer #1.

These included

- Questions re: the appropriateness of the the model system referencing limitations of the cell/cytokine data, lack of cell replenishment, issues of clarification of the status of naive vs LBTI simulations
- Appropriateness of statistical models used
- Minor changes in the manuscript for clarity and conciseness

The authors have answered the questions point by point, revising where needed, adding additional information and citations to bring additional context where needed, and in some cases deciding to keep the text "as is" while explaining their thinking process.

The manuscript has been greatly improved from both the initial review and the response from the authors.

Resubmission Review Comments 12/13/24

Reviewer #1 (Comments for the Author):

All major and almost all minor concerns have been addressed.

One minor issue that remains is that the authors state that they have substituted t-tests in Fig 2, 3, and 4 for the A-measure of stochastic superiority, yet the figure legend for Figure 3 still refers to t-test results.

Response: Thank you for pointing out this inconsistency. We have now updated the Figure 3 caption to reflect the usage of the A-measure of stochastic superiority for statistical comparisons instead of the t-test.

Original Reviewer Comments 8/13/24

Reviewer #1 (Comments for the Author):

Hoerter et al examine how early interactions between human CD4+ T cells and macrophages impact bacterial control during Mycobacterium tuberculosis infection. Specifically, they seek to understand the mechanisms by which LTBI leads to protection from subsequent infection. The authors develop an Agent Based computational model to simulate the cellular and molecular mechanisms that contribute to immune protection within granuloma-like structures. They first use data from published experimental studies (Guirado et al) to calibrate the range of bacterial growth, total cell numbers, and relative frequency of intracellular and extracellular bacteria to their model. Next, they use their model to simulate the early interactions that drive later outcomes for both LTBI and Mtb-naïve samples.

They find that LTBI conditions lead to greater Mtb killing due to more activated infected macrophages, activated CD4+ T cells, and macrophage cell death from activation, while naïve conditions lead to a higher number of macrophage death due to high bacterial load and bursting. The authors also interrogate the frequency of different cell populations within the simulated granuloma-like structures as well as the spatial relationship between activated cells and cytokines TNF α and IFN γ . Overall, the authors find that the granulomas from LTBI samples have more activated CD4+ T cells, less total Mtb, and are more compact than ones from naïve samples. Lastly, the authors identify parameters related to both IFN γ and TNF α secretion and diffusion rates within the ABM that differ between LTBI and naïve samples.

By interrogating how the earliest cellular and molecular interactions between infected macrophages and CD4+ T cells contribute to subsequent disease outcomes, the authors tackle a very important and challenging topic in the field, which is very difficult to address by other approaches. By simulating these interactions, the authors are able to isolate small details about the infection dynamics that they predict have big downstream effects and may potentially point to new therapeutic targets. The power of the model comes from the large number of both host and pathogen variables included and the complexity of their interaction values. Wherever possible, the authors have used starting values that are experimentally derived.

However, there are several limitations to the study and its interpretation that limit enthusiasm for the manuscript in its current form, as described below.

Major Comments:

1) The authors state that the computational model provides new details about the differences in the early interactions between macrophages and T cells between LTBI and naïve samples. While the authors provide substantial details about the different variables that go into the ABM and how they relate to one another, there is little information about which starting and experimental values differ between LTBI and naïve samples. For example, Table 1 provides the initial condition values and ranges, but it's not clear whether these differ between the LTBI and naïve conditions, based on experimental data. In section 3.1, the authors state "Our ABM, calibrated independently to LTBI and naïve experimental group data" and in section 3.4.5., they state the parameters are "stochastically different between LTBI and naïve simulations". The differences in the starting values between LTBI and naïve simulations seems to be critical for deciphering the results of the simulations and understanding other conditions when they might be applicable. This is especially true given that the authors point out counterintuitive conditions such as LTBI simulations "with fewer overall Mtb specific CD4+ T cells." Which starting conditions were experimentally derived and which are significantly different between LTBI and naïve simulations? How do these starting values contribute to the differences in early dynamics that are observed in the simulations?

Response: We would like to address this comment in subsections to ensure that we address all of the individual concerns. **Line numbers refer to the compiled .pdf produced in mSystems.**

- *“there is little information about which starting and experimental values differ between LTBI and naïve samples”*

We have added text (line 283-289) to clarify which experimental data are available in the context of initial conditions. All other available (non-initial condition) experimental data, and their differences, are included in the data that we use to calibrate the model (Figure 2a and 2b).

- *“Table 1 provides the initial condition values and ranges, but it's not clear whether these differ between the LTBI and naïve conditions, based on experimental data”*

We now clarify in the Table 1 caption as well as in the text (lines 283-289) that these values are the same between LTBI and Naïve simulations, and that this assumption is in line with the experimental approach.

- *“In section 3.1, the authors state "Our ABM, calibrated independently to LTBI and naïve experimental group data" and in section 3.4.5., they state the parameters are "stochastically different between LTBI and naïve simulations". The differences in the starting values between LTBI and naïve simulations seems to be critical for deciphering the results of the simulations and understanding other conditions when they might be applicable. This is especially true given that the authors point out*

counterintuitive conditions such as LTBI simulations "with fewer overall Mtb specific CD4+ T cells." Which starting conditions were experimentally derived and which are significantly different between LTBI and naïve simulations? How do these starting values contribute to the differences in early dynamics that are observed in the simulations?"

Hopefully the added clarifications described above show that some initial conditions are set to be the same between LTBI and naïve (these are listed in Table 1) because this is what the experiments did, and what we have experimental data for. The only other initial condition parameters that were varied during calibration (*fractionCD3*, *fractionCD4*, and *fractionTBSpecific*) did not have any available experimental data, and are therefore fitted during calibration. Significant emergent differences in these three parameters (*fractionCD3*, *fractionCD4*, and *fractionTBSpecific*) are shown in Figure 6; and their impacts on the differences in early dynamics are discussed in section 3.4.5.

2) It's unclear why the authors treat TNF as a chemokine and not as a survival or activation signal. While TNF can induce chemokine expression in other cells, TNF itself is not known to act as a chemoattractant. There are many chemokines known to attract different populations of immune cells during Mtb infection, many of which are highlighted in a review cited by the authors (Domingo-Gonzalez et al, PMID: 27763255).

Response: Thank you clarifying this simplifying assumption in the model. In our simulation, TNF α serves as a proxy for generalized chemokine, as well as an activation signal for macrophages which promotes survival of macrophages. Yes, TNF α induces chemokine expression which helps promote cellular movement, but studies show that without TNF α expression, granulomas do not form as noted in the review paper by Domingo-Gonzalez. For the sake of computational efficiency (which is greatly affected by each additional diffusing molecule) we chose to assume a pseudo-steady state relationship between TNF α and its downstream chemokines, thereby allowing us to use TNF α as a proxy for chemokines. We have clarified this assumption in the methods (lines 138-140).

3) There is a lack of clarity around the statistical analysis. The figure legends for Figures 2, 3, and 4 state that every parameter is statistically significant between LTBI and naïve samples apart from a few conditions with "n.s.", but there are many places where LTBI and naïve values look totally identical (especially in Figure 4). Additional details should be provided.

Response: As we noted in Section 2.5, the *t*-test performed for figures 2-4 is overpowered when comparing LTBI and naïve simulation outputs since the number of samples is greater than 2000. However, to alleviate this confusion, we have now (instead of the *t*-tests) performed the A-measure of stochastic superiority on these outputs as was already done for Figure 6, parameter comparisons. The threshold for the scaled A-measure of stochastic superiority is 0.71 (1). Above this threshold, comparisons are considered meaningfully (not just statistically) significant. Additionally, we have added "Refer to Section 2.5 for more details" in figure captions where necessary, referring readers to the methods section.

4) The manuscript would benefit from a more lengthy discussion about the benefits and limitations of the computational model. There are several variables and assumptions that strongly impact the results:

a. The authors chose to model a closed system, which does not account for additional cell recruitment that normally occurs in a lung infection during the formation of a granuloma structure.

Response: The experimental data described in this paper is a closed system with no cell replenishment. There exist models of open systems (2–7). Part of the novelty of this simulation is that it replicates an *in vitro* system to evaluate the earliest dynamics of granuloma formation not long-term dynamics. However, we have added some discussion about the scope of our predictions being currently limited to this closed system context (lines 760-764).

b. The lack of other cell types that are known to contribute to early events during infection and to granulomas.

Response: Although the dominant immune cells are present in the model, we acknowledge there are several other cell types not included in this simulation that may be present in granulomas. We previously addressed this in the limitations section of the discussion, and we now expand this discussion to be more specific (lines 754-756).

c. The use of IFN γ cytokine as 1 of only 2 signals that leads to macrophage activation and bacterial control, despite experimental evidence that IFN γ is not sufficient for control of Mtb.

Response: We completely agree with the reviewer that IFN γ alone is not sufficient for Mtb control and do not believe that we make that claim. In our model, IFN γ plays a role in macrophage activation, and macrophage activation in turn can affect bacterial numbers, but these interactions are only a small part of the overall system of interactions, and indeed not the only drivers. Numerous studies have shown that IFN γ is a critical player in the control of Mtb, especially IFN γ secreted from Mtb-specific CD4 T cells (8–13). In agreement with these findings, we also conclude how important of a factor IFN γ is, but we discuss how it's one of several factors that synergize to help LTBI donor simulations control Mtb better than naïve donor simulations (lines 690-699, 711-713, 765-769).

d. The exclusion of factors that have been identified experimentally that lead to an immunosuppressive environment within a granuloma, including TGF-beta and IDO signaling (McCaffrey E, PMID: 35058616; Gern B et al, PMID: 33711270).

Response: We completely agree that accounting for immunosuppressive or anti-inflammatory mechanisms is important when simulating immune responses. Our model indirectly accounts for feedback between pro- and anti-inflammatory signals through our activation-time parameters (lines 214-216). We believe that this simplifying assumption is warranted because we are investigating such short and early time points. The studies provided regarding TGF β and IDO signaling have added new valuable insight into the immunosuppressive environment within a granuloma, but the timeframe and the type of granuloma these findings were conducted in are quite different than our biological system. The TB granulomas analyzed in McCaffrey E et al, are from tissue samples of active TB cases of unknown length of time and post-mortem tissue samples. The IDO findings from Gern B et al are in mice and NHPs with established granulomas.

Minor Comments:

1) The methods section is lengthy and reads more like a results section. The sentences within the methods section could be condensed down for clarity.

Response: We have carefully reviewed the methods section, and felt that the information included in the methods is necessary for transparency about which mechanisms are explicitly incorporated into the model, and which simplifying assumptions are being made - this need for clarity was also highlighted by the reviewer (Major comments #1 and 3). We are not sure which sections the reviewer is referring to that reads more like a results section, but we assume they are referring to the calibration section where we do discuss some calibration outcomes. Based on this assumption: we feel that this information is best included in the calibration methods section because it is fundamental to describing the calibration process, and numbers pertain to specific elements of the calibration process, which would be repetitive to describe if we moved these to the results section.

2) In the discussion, the authors assert that "two types of immune memory could be represented in our model." It would be helpful to show the starting values for LTBI versus naïve samples that support this point.

Response: We believe that our responses to Major comment #1 and Minor comment #4 should help clarify that our model-predicted differences in initial conditions are reflected in the parameter values described in Figure 6e. Our discussion here of the relationship between these predicted initial conditions and the interpretation as adaptive immune memory, is integrating/synthesizing our findings from Fig 6e for the counterintuitive (normalized) starting values for total cd4s and tb spec cd4s, fig 2b for total cell fold change and fig 3d for activated cd4s. We have added specific references to these figures in the discussion (lines 696, 699, 713,716).

3) The text describes results for Fig 6, which are actually shown in Fig 5.

Response: Figure field codes have been updated and are correct now.

4) How was "Mtb specific CD4+ T cells" versus "non-specific CD4+ T cells" experimentally defined? Does it make sense for naïve samples to start with a population of Mtb-specific CD4+ cells?

Response: We believe that our responses to Major Comment #1 should address the first part of this question. We do not have experimental data for this metric, and therefore estimate these values through calibration. As for the assumption that naïve samples have some Mtb specific cells, we assume that these T cells are naïve (i.e. have not encountered their cognate antigen yet, and are therefore not effector cells) and are part of the diverse repertoire of TCRs capable of responding to novel antigens throughout a human's life (14). We've added lines 279-281 to clarify this.

References cited in this response to reviewers:

1. Hamis S, Stratiev S, Powathil GG. Uncertainty and sensitivity analyses methods for agent-based mathematical models: An introductory review. *Physics Of Cancer, The: Research Advances*. 2020;1–37.
2. Segovia-Juarez JL, Ganguli S, Kirschner D. Identifying control mechanisms of granuloma formation during *M. tuberculosis* infection using an agent-based model. *J Theor Biol* [Internet]. 2004 Dec;231(3):357–76. Available from: <https://linkinghub.elsevier.com/retrieve/pii/S0022519304003212>
3. Ray JCJ, Flynn JL, Kirschner DE. Synergy between Individual TNF-Dependent Functions Determines Granuloma Performance for Controlling *Mycobacterium tuberculosis* Infection. *The Journal of Immunology* [Internet]. 2009 Mar 15;182(6):3706–17. Available from: <https://journals.aai.org/jimmunol/article/182/6/3706/103675/Synergy-between-Individual-TNF-Dependent-Functions>
4. Fallahi-Sichani M, El-Kebir M, Marino S, Kirschner DE, Linderman JJ. Multiscale Computational Modeling Reveals a Critical Role for TNF- α Receptor 1 Dynamics in Tuberculosis Granuloma Formation. *The Journal of Immunology*. 2011;186(6):3472–83.
5. Cicchese JM, Evans S, Hult C, Joslyn LR, Wessler T, Millar JA, et al. Dynamic balance of pro- and anti-inflammatory signals controls disease and limits pathology. *Immunol Rev* [Internet]. 2018 Sep 11;285(1):147–67. Available from: <https://onlinelibrary.wiley.com/doi/10.1111/imr.12671>
6. Pienaar E, Cilfone NA, Lin PL, Dartois V, Mattila JT, Butler JR, et al. A computational tool integrating host immunity with antibiotic dynamics to study tuberculosis treatment. *J Theor Biol* [Internet]. 2015 Feb;367:166–79. Available from: <https://linkinghub.elsevier.com/retrieve/pii/S0022519314006730>
7. Warsinske HC, Pienaar E, Linderman JJ, Mattila JT, Kirschner DE. Deletion of TGF- β 1 increases bacterial clearance by cytotoxic t cells in a tuberculosis granuloma model. *Front Immunol*. 2017;8(DEC).
8. Green AM, DiFazio R, Flynn JL. IFN- γ from CD4 T Cells Is Essential for Host Survival and Enhances CD8 T Cell Function during *Mycobacterium tuberculosis* Infection . *The Journal of Immunology*. 2013;190(1):270–7.
9. Flynn JL, Chan J, Triebold KJ, Dalton DK, Stewart TA, Bloom BR. An essential role for interferon gamma in resistance to *Mycobacterium tuberculosis* infection. *J Exp Med* [Internet]. 1993 Dec 1;178(6):2249–54. Available from: <https://rupress.org/jem/article/178/6/2249/25121/An-essential-role-for-interferon-gamma-in>
10. Gern BH, Adams KN, Plumlee CR, Estes JD, Urdahl KB, Gerner MY, et al. Article TGF b restricts expansion , survival , and function of T cells within the tuberculous granuloma Article TGF b restricts expansion , survival , and function of T cells within the tuberculous granuloma. 2021;1–13.
11. Domingo-Gonzalez R, Prince O, Cooper A, Khader SA. Cytokines and Chemokines in *Mycobacterium tuberculosis* Infection. In: *Tuberculosis and the Tubercle Bacillus* [Internet]. Washington, DC, USA: ASM Press; 2017. p. 33–72. Available from: <http://doi.wiley.com/10.1128/9781555819569.ch2>
12. Mai D, Jahn A, Murray T, Morikubo M, Lim PN, Cervantes MM, et al. Exposure to *Mycobacterium* remodels alveolar macrophages and the early innate response to *Mycobacterium tuberculosis* infection. Salgame P, editor. *PLoS Pathog* [Internet]. 2024 Jan 18;20(1):e1011871. Available from: <http://dx.doi.org/10.1371/journal.ppat.1011871>

13. Nemeth J, Olson GS, Rothchild AC, Jahn AN, Mai D, Duffy FJ, et al. Contained Mycobacterium tuberculosis infection induces concomitant and heterologous protection. Sassetti CM, editor. PLoS Pathog [Internet]. 2020 Jul 16;16(7):e1008655. Available from: <http://dx.doi.org/10.1371/journal.ppat.1008655>
14. Kumar B V., Connors TJ, Farber DL. Human T Cell Development, Localization, and Function throughout Life. Immunity [Internet]. 2018 Feb;48(2):202–13. Available from: <https://doi.org/10.1016/j.immuni.2018.01.007>

Re: mSystems01290-24R1 (Timing matters in Macrophage / CD4+ T cell interactions: An agent-based model comparing *Mycobacterium tuberculosis* host-pathogen interactions between latently infected and naïve individuals)

Dear Dr. Elsje Pienaar:

Your manuscript has been accepted, and I am forwarding it to the ASM production staff for publication. Your paper will first be checked to make sure all elements meet the technical requirements. ASM staff will contact you if anything needs to be revised before copyediting and production can begin. Otherwise, you will be notified when your proofs are ready to be viewed.

Sincerely,
Andrew Bartko
Editor
mSystems

Reviewer #1 (Comments for the Author):

All minor concerns have been addressed.